# ROYAL SOCIETY
# OPEN SCIENCE

**Subject Category:**
Biology (whole organism)

evolution/biomechanics

bone microanatomy, evolutionary transition, locomotion, long bones, mustelidae, trabecular bone architecture

**Author for correspondence:**
E. Amson
e-mail: eli.amson@mfn.berlin

# Trabecular bone architecture in the stylopod epiphyses of mustelids (Mammalia, Carnivora)

## E. Amson and B. M. Kilbourne

Museum für Naturkunde, Leibniz-Institut für Evolutions- und Biodiversitätsforschung, Berlin, Germany

EA, 0000-0003-1474-9613; BMK, 0000-0002-5940-0821

Mustelidae, a carnivoran clade that includes for instance weasels, badgers, otters and martens, has undergone several evolutionary transitions of lifestyle, resulting in specializations for fossorial, natatorial and scansorial locomotion, in addition to more generalized species. The family is therefore regarded as offering an adequate framework for morpho-functional analyses. However, the architecture of the epiphyseal trabecular bone, which is argued to be particularly responsive to the biomechanical environment, has never been studied. Here, we quantify trabecular bone parameters of the proximal and distal epiphyses of the humerus and femur in 29 species of mustelids and assess the differences of these parameters among groups defined *a priori* based on the aforementioned locomotor types. The parameters are assessed in a phylogenetic framework, taking into account the potential effect on an individual's body mass. The range of variation described by the acquired parameters is relatively restricted when compared to that of other clades. Generalists, however, are featuring a wider range of variation than the other types. While clear discrimination of locomotor types is difficult, some differences were highlighted by our analysis, such as a greater bone fraction associated with the natatorial taxa, which we discuss in a functional context.

## 1. Introduction

Mammals occur on all continents and in all biomes [1], and one key to their evolutionary success has been morphological specializations to meet the biomechanical demands of their ecological niches. As such, the form–function relationships of the mammalian locomotor system have been the subject of study for well over a century (e.g. [2]). In more recent years, the discovery of fossil mammaliaform taxa with clear specializations for locomotion (e.g. [3–5]) has further underscored that specialized limb morphology

has been an integral aspect of mammalian evolution stretching back to the clade's infancy [6]. That mammals have diversified to meet the biomechanical demands of a wide range of ecological niches, as well as the early appearance of mammaliaform locomotor specializations, stresses that the study of locomotor morphology is vital to our understanding of mammalian ecology and evolution.

The study of form–function relationships and locomotor morphology, particularly in a phylogenetic context, requires a lineage of mammals displaying a diversity of functional specializations. Mustelidae are a carnivoran lineage exhibiting a diversity of locomotor habits. Specifically among mustelids, three main locomotor specializations can be defined [7,8]: badgers are fossorial, featuring specialized digging skills; martens are scansorial, as they are preponderantly found climbing; otters are natatorial, being skilful swimmers. In addition to these three specializations, weasels exhibit a more generalized locomotor habit. In spite of a flurry of recent studies, mustelids have been used as an ideal group to understand locomotor form–function relationships and their evolution for nearly four decades [7–12]. Though the majority of these studies have focused on long bone gross morphology [8–11,13–18], more recent studies have started to investigate cross-sectional properties and microanatomy [8,19].

Bone trabeculae are microanatomical struts that dynamically respond to strain incurred from the mechanical loads that the bone experiences [20–23]. Thus, trabecular traits should probably vary according to locomotor habits that are distinguished by differing biomechanical demands. Among trabecular traits, the degree of anisotropy (DA) indicates the degree to which trabeculae are oriented in a single direction. Bone fraction (BV/TV)—bone volume relative to total volume—reflects the bone fraction of the studied volume of interest (VOI). Connectivity (Conn.D) approximately denotes the number of trabeculae per unit volume, whereas trabecular thickness (Tb.Th) and separation (Tb.Sp), respectively, denote the thickness of trabecular struts and the separation among them. The bone surface area (BS) refers to the external surface of the trabecular network within the VOI.

Given that differing locomotor types are distinguished by differences in the gross morphology of limb bones (e.g. [7,9,16,18]) and their cross-sectional traits [8], we expect that trabecular traits also distinguish mustelid locomotor types. The present analysis uses humeral and femoral data to detect potential differences in forelimb and hindlimb bone structure relating to locomotor differences. During digging, the forelimbs of fossorial mustelids function in soil, the high density of which (1.83–2.85 g cm$^{-3}$ [24]) probably subjects the forelimbs to high mechanical loads and consequently greater mechanical strains. Therefore, we predict that DA and BV/TV will be greater in the forelimbs of fossorial taxa compared to non-fossorial mustelids; greater values of BV/TV could in turn entail greater values of Tb.Th and Conn.D and lower values of Tb.Sp that could act to withstand these heightened strains. The investigation of xenarthran trabecular bone architecture of the forelimb did suggest that high values of DA, in particular, could be associated with strenuous activities such as digging [25]. A tendency for higher Conn.D values was recovered for fossorial sciuromorphs' femoral head [26]. Natatorial mustelids function in water, which is also highly dense (1.0 g cm$^{-3}$), suggesting that their limb bones may also need to resist high mechanical loads when paddling with the limbs [27]. However, recent studies have shown that the need for tetrapods swimming in shallow water to counteract buoyancy is probably a more critical factor than resisting loads incurred during the paddling of limbs [28,29]. We thus predict that all investigated VOIs of natatorial mustelids will have greater values of BV/TV, Tb.Th and Conn.D and lower values of Tb.Sp than mustelids of other locomotor habits so as to counteract positive buoyancy. This differs from our expectations regarding fossorial taxa, for which only the forelimb VOIs are expected to follow this trend. While we do not have a specific prediction regarding BS, we also included this parameter in our analysis, because it was found in previous analyses as discriminating functional groups (e.g. [30,31]; therein a ratio to BV and/or TV)

In this study, we quantify stylopod long bone trabecular properties in mustelids and compare them to those of previous studies on the trabecular architecture of other clades. Such an approach aims at better understanding the functional implications of trabecular bone architecture in a comparative framework.

# 2. Material and methods

## 2.1. Specimens and scanning procedure

To measure the trabecular traits of the humerus and femur in mustelids, we sampled 35 individuals across 29 species (table 1 and figure 1). Humeri and femora were scanned at the Museum für Naturkunde Berlin using a Phoenix|x-ray Nanotom (GE Sensing and Inspection Technologies GmbH, Wunstorf, Germany), at the Royal Veterinary College (Hertfordshire, UK) using a GE LightSpeed

**Table 1.** Sampled mustelid species with their locomotor types. Scan location indicates the facility where CT scans were made: (1) Museum für Naturkunde Berlin, Berlin, Germany, (2) Royal Veterinary College, Hertfordshire, UK and (3) University of Chicago, Chicago, USA.

| species | common name | habit | N | scan location |
|---|---|---|---|---|
| *Amblonyx cinereus* | Asian small-clawed otter | natatorial | 1 | 1 |
| *Arctonyx collaris* | Hog badger | fossorial | 1 | 1 |
| *Eira barbara* | Tayra | scansorial | 1 | 1 |
| *Enhydra lutris* | Sea otter | natatorial | 1 | 1 |
| *Galictis vittata* | Greater grison | generalized | 1 | 3 |
| *Gulo gulo* | Wolverine | generalized | 2 | 1,2 |
| *Ictonyx libyca* | Saharan striped polecat | fossorial | 1 | 1 |
| *Ictonyx striatus* | Striped polecat | fossorial | 2 | 1 |
| *Lontra longicaudis* | Long-tailed otter | natatorial | 1 | 1 |
| *Lutra lutra* | Eurasian otter | natatorial | 1 | 1 |
| *Lutrogale perspicillata* | Smooth-coated otter | natatorial | 1 | 3 |
| *Martes americana* | N. American marten | scansorial | 1 | 3 |
| *Martes flavigula* | Yellow-throated marten | scansorial | 2 | 1 |
| *Martes foina* | Beech marten | scansorial | 1 | 1 |
| *Martes martes* | Pine marten | scansorial | 2 | 1 |
| *Martes zibellina* | Sable | scansorial | 2 | 1 |
| *Meles meles* | European badger | fossorial | 1 | 1 |
| *Mellivora capensis* | Honey badger | fossorial | 1 | 1 |
| *Melogale moschata* | Chinese ferret-badger | fossorial | 1 | 1 |
| *Mustela erminea* | Ermine | generalized | 1 | 1 |
| *Mustela eversmanii* | Steppe polecat | generalized | 1 | 1 |
| *Mustela itatsi* | Japanese weasel | generalized | 1 | 1 |
| *Mustela kathiah* | Yellow-bellied weasel | generalized | 1 | 1 |
| *Mustela lutreola* | European mink | natatorial | 1 | 1 |
| *Mustela putorius* | European polecat | generalized | 1 | 1 |
| *Mustela sibirica* | Siberian weasel | generalized | 1 | 1 |
| *Pekania pennanti* | Fisher | scansorial | 1 | 1 |
| *Taxidea taxus* | N. American badger | fossorial | 2 | 1 |
| *Vormela peregusna* | Marbled polecat | generalized | 1 | 1 |

16 scanner (GE Medical Systems, Pollards Wood, UK) and at the University of Chicago using a Phoenix | X-ray Nanotom and v | tome | x combination, with a resolution ranging between 17 and 34 µm. The scans were then processed with the software VG Studio Max 2.0 and 2.1 (Volume Graphics, Heidelberg, Germany). In order to assess whether the employed resolution was sufficient for trabecular architecture analysis, we made sure that the relative resolution (mean trabecular thickness divided by resolution; [33]) was not lower than 5 pixels/trabecula [26,33,34]. Right bones were used when possible and, if not, the left bone was used and mirrored. Acquired parameters from each individual specimen are provided in electronic supplementary material, S1.

## 2.2. Trabecular parameters acquisition

Each bone was manually oriented in VG Studio Max to have its proximodistal axis aligned with the vertical axis of the three-dimensional coordinate space employed by VG Studio and its anterior face oriented to the left of the image. Each bone was then exported as a DICOM stack of greater

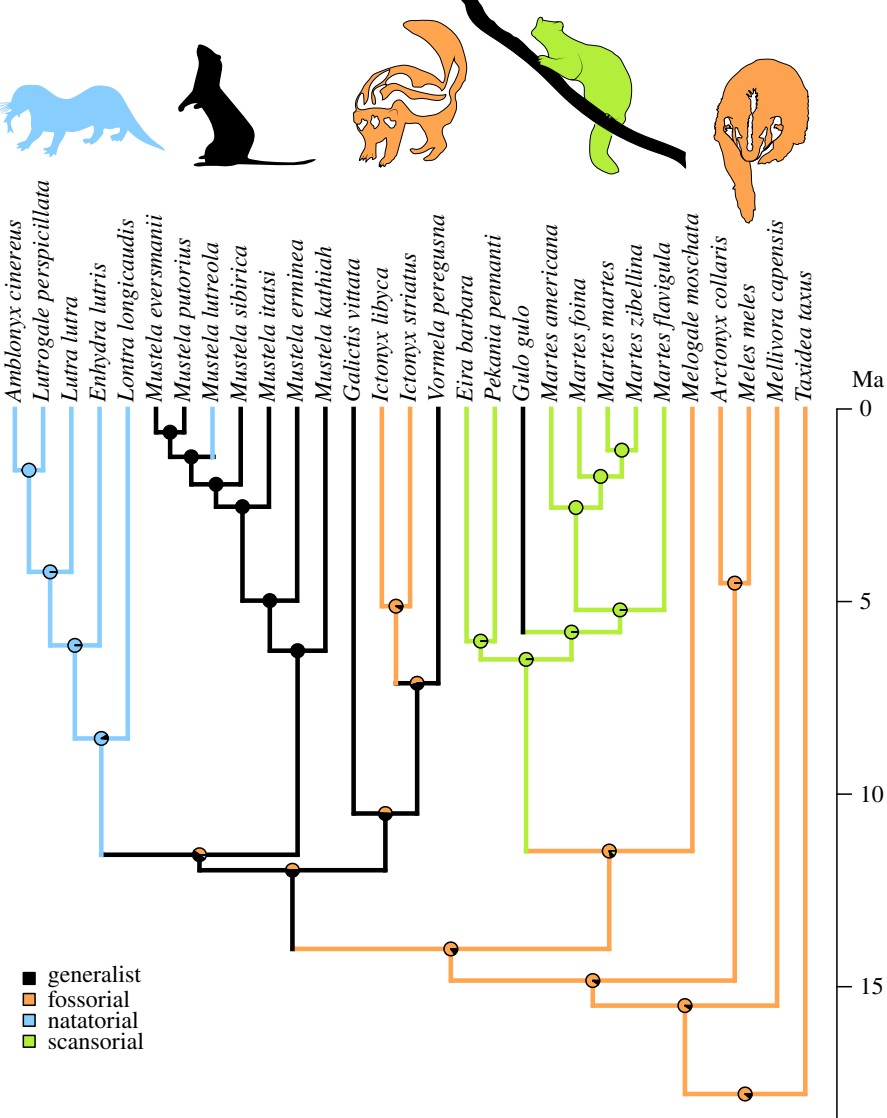

**Figure 1.** Lifestyle transitions represented by the sampled mustelids. Branch painting represents the history of the four lifestyles according to a reconstruction using parsimony (no ambiguous state was recovered for any of the nodes). Pie charts at the nodes represent the result of the stochastic mapping of the character (see Material and methods for more details). Timetree topology and branch lengths are from Law *et al.* [32].

than 1000 image slices, with the image stacks having been generated sequentially along the bone's proximodistal axis.

Measurements of trabecular parameters were taken on substacks corresponding to cubic VOIs. The substacks were defined using a custom macro (electronic supplementary material, S2) for the Fiji package (ImageJ2 and plugins [35–37]), which relies on the ImageJ plugin TransformJ [38]. This macro was used to extract a substack centred in each studied articular structure—i.e. the humeral head, humeral trochlea, femoral head and lateral condyle of the femur—that is as big as possible without including cortical bone. These VOIs were chosen in order to compare the trabecular architecture of mustelids to that of other clades similarly studied [25,26,39]. The articular structures of some small-sized mustelids only contained a few trabeculae (see the over-sampling problem described by Fajardo & Müller [40] for dataset including a large size range). We therefore chose this approach to acquire, for each VOI, a relatively large amount of the trabeculae of the centre of the studied articular structure, which can be viewed as functionally analogous across the dataset. We consider this criterion as necessary to define the size and location of the VOIs across our dataset, of which the definition was shown to drastically influence some trabecular parameters [34]. One should note, however, that this criterion may entail different proportions of the subarticular trabeculae sampled among the VOI

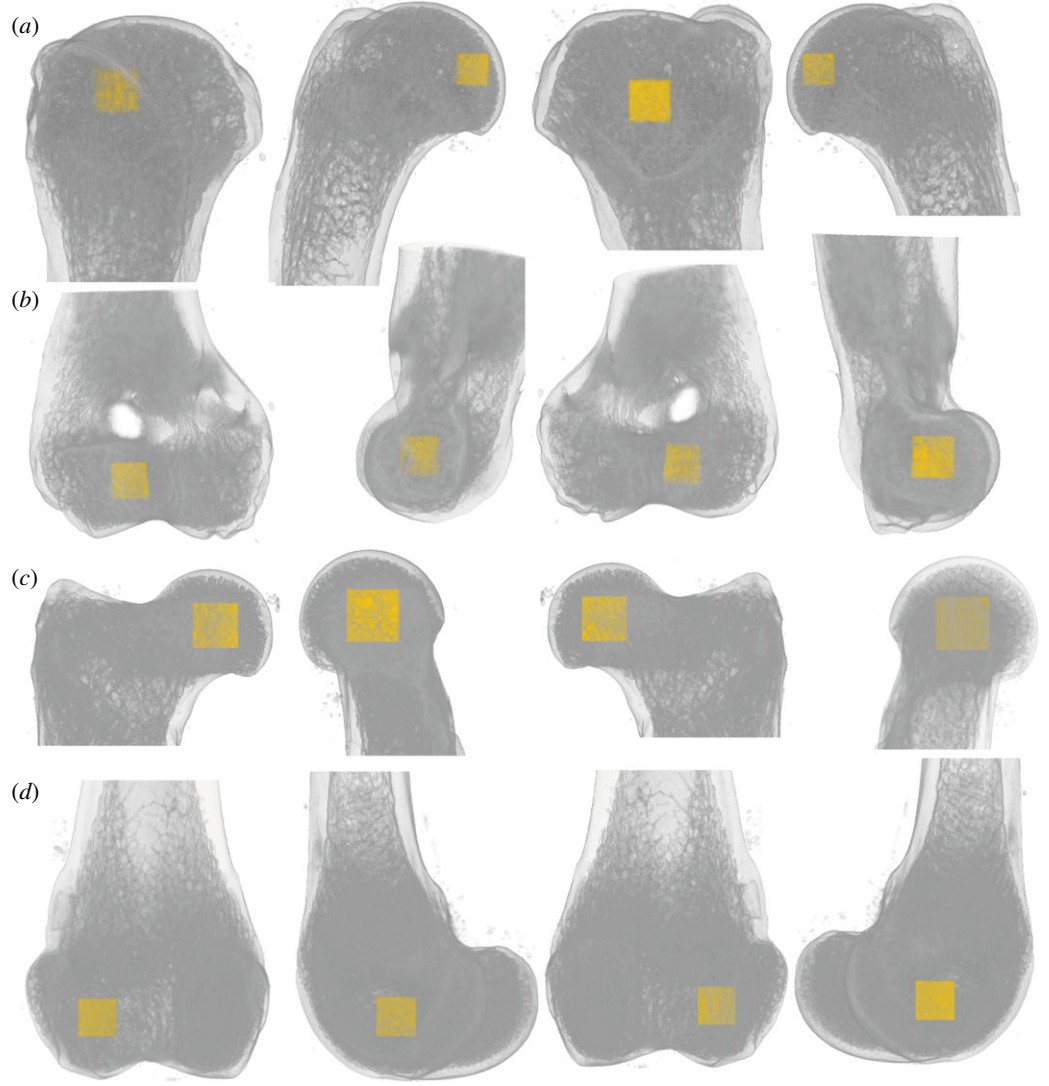

**Figure 2.** Definition of the volumes of interest (VOIs) used to quantify trabecular architecture in the mustelid humeral and femoral epiphyses. (*a*) Humeral head; (*b*) humeral trochlea; (*c*) femoral head; (*d*) femoral lateral condyle; from left to right: anterior, medial, posterior, lateral views. Example specimen: *Amblonyx cinereus*, ZMB_MAM 43245 (Asian small-clawed otter). See also electronic supplementary material, S3 for smallest and largest taxa of the dataset.

types. This can also entail for some ROIs to comprise a greater proportion of the trabeculae directly internal to the articular surface (and for other ROIs to encompass relatively more trabeculae more distant to the surface). Both issues can yield a potential bias when comparing different VOI types (see inter-limb ratios below). Given the different morphologies of the humeral and femoral articulation in mustelids, we do not consider that avoiding such a bias is achievable. To define the centre of each articular structure, first, the proximodistal middle was defined, as the mid-level between the proximal-most and distal-most levels of the corresponding articular surface. Then, the central point at this level was defined for each VOI using the anterior, posterior, medial and lateral edges of the corresponding articular surface (see the resulting selection in figure 2 and electronic supplementary material, S3). The humeral head VOI was bounded anterolaterally by the maximum concavity of the lateral side of head, medially by medial-most point at the level of the anterolateral corner (just defined) and posteriorly by the posterior-most level of head (note that the VOI does not appear as centred in figure 2*b,d*, because the anteroposterior depth of the articular surface at mid-length does not extend as far anteriorly as the anterior-most edge of the surface visible in the figure). The humeral trochlea VOI was bounded laterally by the anterolateral corner of the trochlea, medially and anteriorly by the anteromedial corner of the trochlea and posteriorly by the posterior-most level of trochlea at the mediolateral centre of the VOI (just defined). The femoral head VOI was bounded anteriorly, medially and posteriorly by

the border of articular surface and laterally by most concave point just lateral to the posterior side of articular surface. The lateral condyle VOI was bounded anteriorly by the patellar surface, posteriorly by condyle articular surface, medially by the most concave point of the intercondylar notch and laterally by the lateral-most level of the bone surface. All VOIs just defined were similarly defined in previous trabecular bone analyses (e.g. [39,41]) except for that of the humeral trochlea. Other distal humerus VOIs could have been used, such as one related to the capitulum, but inspection of the scans at hand revealed that too few trabeculae would be included in such VOIs, especially for the smaller taxa.

We assessed the intra-observer error by repeating 10 times in a row the acquisition of the humeral head VOI for one specimen (ZMB_MAM 60513, *Gulo gulo*). The coefficients of variation of the acquired parameters ranging from 0.14 to 4.71% (mean of 2.19%; electronic supplementary material, S4), we consider that this methodology is fairly robust to observer error.

Using the BoneJ plugin [42], the extracted substacks were thresholded (i.e. foreground colour is associated with bony tissue, and background colour with anything else; 'Optimize Threshold > Threshold Only' routine), purified (i.e. isolated particles are removed; 'Purify' routine) and were analysed for the trabecular parameters: DA (no units), $V_{1,1}$, $V_{2,1}$ and $V_{3,1}$, respectively, the $x$-, $y$- and $z$-components of the eigenvector defining the main orientation of the anisotropy (no units), connectivity (used for the scan quality assessment only; roughly the number of trabeculae) and connectivity density (Conn.D; number per mm$^3$), BV (mm$^3$), total volume of the VOI (TV; mm$^3$), trabecular mean thickness (Tb.Th; mm), trabecular mean separation (Tb.Sp; mm) and BS (mm$^2$). The main direction of the trabeculae (MDT; no units), acquired for descriptive purposes and shown in electronic supplementary material, S5, was derived from $V_{1,1}$, $V_{2,1}$ and $V_{3,1}$ with the function *cosap* of R package Directional [43].

In addition to the trabecular parameters corresponding to each VOI, we also compared inter-limb ratios, i.e. ratios between the proximal humerus and proximal femur VOIs' values, and distal humerus and distal femur VOIs' values, respectively.

## 2.3. Specimens' body mass estimation

All computations were performed with R v. 3.5.1 [44]. As a body size proxy, we used the specimens' body mass estimation (BMsp; because actual body mass is unknown for most collection specimens). This estimation was obtained, for each VOI, using a regression of the species mean body mass (the latter is taken from the global database of late Quaternary mammal, MOM v. 4.1; [45]; unit: g) against a measure taken directly on each specimen, TV (see above). TV was chosen because it directly relates to the size of the articular structure under study, being defined as the largest cube fitting in its centre (see VOI definitions above). Other studies use linear measurements such as femoral head radius [39] or height [46] to represent the individual's size. While we did not quantify the difference yielded by using these alternative metrics, we assume that our approach should be at least as accurate. Indeed, the diameter of two epiphyses of the same overall size can strongly differ according to the epiphysis' shape). Future endeavours to estimate mustelid body mass based on skeletal measurements should include a dataset of specimens with known body mass. This was unfortunately not feasible for the present study. The regression was phylogenetically informed using the *lm.rrpp* function (RRPP package [47]) and a within-group correlation structure based on the optimized Pagel's lambda value [48]. Lambda was recovered using the functions *gls* (nlme package, [49]) and *corPagel* (ape package, [50]) and the timetree taken from [32] (figure 1). This timetree's branch lengths were scaled according to the recovered lambda value for each VOI with the *rescale* function (Geiger package, [51]). It is this transformed tree that was used for rrpp regression. If the recovered lambda was negative or greater than 1, its value was forced to be 0 or 1, respectively. Finally, the estimation of the BMsp (unit: natural logarithm of mass in g) was obtained based on the fit of the latter regression using the *predict* function (base of R).

## 2.4. Locomotor types and body size

The general procedure aims to describe the variation of trabecular parameters in mustelids in relation to their locomotor types. For all linear models fitted for this purpose (i.e. ANOVA and ANCOVA), we adopted a phylogenetically informed approach using the *lm.rrpp* function and a within-group correlation structure based on the optimized Pagel's lambda value (see above). Following Kilbourne [7] and Kilbourne & Hutchinson [8], we recognize four main locomotor types: generalist, fossorial, natatorial and scansorial

(table 1). As body size is correlated with some, if not all, trabecular parameters [39,46], we first tested in our dataset for the presence of such a correlation with a linear regression of each parameter on body size proxy (BMsp, see above). In the case of a significant correlation with body size, we used an ANCOVA, with BMsp as a covariate to assess each parameter's difference among locomotor types (and differences were visualized with boxplots showing the residuals of the linear regression). Otherwise, an ANOVA was used. A phylogenetic principal components (pPCs) analysis (*phyl.pca* function, phytools package [52]) was used to represent the data in a multivariate fashion. Because the locomotor types did not significantly differ in their BMsp (as indicated by a phylogenetically informed ANOVA performed as described above; $p$-value = 0.095, $F$-value = 2.399, $R^2$ = 0.224), it should be possible to exclude the effect of size from that of the locomotor type. But one should note that the sampled generalist mustelids tend to be smaller than the members of the other locomotor types (electronic supplementary material, S6).

After mapping the locomotor types onto the terminal branches of the mustelid phylogeny [32], we reconstructed locomotor type transitions along the internal branches of the mustelid phylogeny by performing ancestral state reconstruction using parsimony (*ancestral.pars* function, phangorn package; [53]) and stochastic character mapping (relying on a maximum-likelihood estimation using equal-rates model and 1000 simulations; *make.simmap* function, phytools package [52]). In order to assess the correlation between the sampled phylogeny and the locomotor types, we used two-block partial least-squares analysis assuming a Brownian motion model of evolution [54] and the retention index (RI; *RI* function of phangorn package; [53]).

# 3. Results

Ancestral state reconstruction of locomotor type finds that a fossorial type is the most likely ancestral condition for the clade, indicating that there have been five transitions along the phylogeny's internal branches to the three non-fossorial types (generalist, natatorial and scansorial), and one reversal to a fossorial type (figure 1). The correlation between the phylogeny and the locomotor types as they are distributed among the phylogeny's terminal branches is very strong (two-block partial least-squares analysis, rPLS = 0.87; $p < 0.0001$; RI = 0.82), indicating that the sampled representatives of each locomotor type are highly aggregated (i.e. clustered) within the phylogeny. One has to keep in mind that a strong aggregation of locomotor type within the phylogeny poses a statistical challenge [54] when trying to understand the covariation [54] between trabecular parameters and locomotor types. Nevertheless, the chosen procedure is considered as robust under group aggregation [54], even though the possible correlations uncovered by our analysis mostly relate to singular ecological shifts.

A mapping of the trabecular parameters of interest upon the phylogeny is provided in electronic supplementary material, S6. Descriptive statistics for each VOI and each parameter of interest are provided in table 2 (for proximal VOIs) and electronic supplementary material, S7 (for all VOIs).

For the humeral head VOI (figure 3 and table 3), DA is the only parameter that does not significantly differ among locomotor types. Conn.D is significantly greater in fossors and generalists than in the other two locomotor types (Fossorial:Natatorial, $p$-value = 0.008; Fossorial:Scansorial, $p$-value = 0.003; Generalist:Natatorial, $p$-value = 0.003; Generalist:Scansorial, $p$-value = 0.002). With regard to Tb.Th, fossorial and generalist taxa both have significantly lower values than natatorial and scansorial taxa, with four groups forming a gradient (figure 3; Fossorial:Natatorial, $p$-value = 0.003; Fossorial:Scansorial, $p$-value = 0.003; Generalist:Natatorial, $p$-value = 0.003; Generalist:Scansorial, $p$-value = 0.003). BV/TV values are significantly greater in generalists and natators than in fossors (scansors tend to be intermediate). For Tb.Sp, fossorial and generalist taxa have significantly lower values than scansorial taxa (Fossorial:Scansorial, $p$-value = 0.016; Generalist:Scansorial, $p$-value = 0.007). Even though there is an overall difference in BS among the types (i.e. ANCOVA's $p$-value is significant for that trait), pairwise comparisons reveal that differences are not significant.

For the humeral trochlea VOI (electronic supplementary material S8 and S9), DA and BS only differ significantly between fossors and generalists among the four locomotor types (DA, $p$-value = 0.002; Fossorial:Generalist, $p$-value = 0.035), with these two groups exhibiting lower and higher values, respectively. BV/TV tends to be higher in natators, but not significantly so. With regard to Tb.Th, most fossorial taxa have lower values (but they only differ significantly from the natators and scansors, $p$-values = 0.007 and 0.021, respectively). Natators and scansors further exhibit higher Tb.Th values than generalists (Generalist:Natatorial, $p$-value = 0.004; Generalist:Scansorial, $p$-value = 0.015). Neither Conn.D nor Tb.Sp conspicuously differs among the locomotor types.

**Table 2.** Mean, s.d. and range of trabecular parameters of interest for each mustelid locomotor type. Data for the humeral head, femoral head and ratio between these two volumes of interest (VOIs) are presented. See electronic supplementary material, S7 for all studied VOIs.

| | fossorial | generalist | natatorial | scansorial | all |
|---|---|---|---|---|---|
| **humeral head** | | | | | |
| DA | 0.57 (0.1) 0.45–0.71 | 0.56 (0.12) 0.42–0.77 | 0.54 (0.11) 0.37–0.66 | 0.5 (0.08) 0.34–0.58 | 0.54 (0.1) 0.34–0.77 |
| Conn.D | 33.87 (20.04) 13.79–74.81 | 30.61 (12.13) 11.32–46.75 | 18.21 (7.37) 8.91–25.6 | 16.86 (7.21) 11.66–32.63 | 25.51 (14.31) 8.91–74.81 |
| BV/TV | 0.34 (0.04) 0.28–0.4 | 0.4 (0.06) 0.27–0.46 | 0.41 (0.03) 0.38–0.47 | 0.37 (0.02) 0.33–0.4 | 0.38 (0.05) 0.27–0.47 |
| Tb.Th.Mean | 0.14 (0.02) 0.12–0.18 | 0.17 (0.01) 0.15–0.19 | 0.19 (0.02) 0.16–0.21 | 0.18 (0.02) 0.15–0.2 | 0.17 (0.02) 0.12–0.21 |
| Tb.Sp.Mean | 0.33 (0.05) 0.25–0.39 | 0.31 (0.08) 0.26–0.46 | 0.36 (0.03) 0.33–0.42 | 0.4 (0.06) 0.31–0.47 | 0.35 (0.07) 0.25–0.47 |
| BS | 578.46 (690.46) 18.45–1902.83 | 162.94 (367.37) 10.51–1137.21 | 503.99 (582.37) 50.56–1632.58 | 157.21 (126.96) 45.6–409.84 | 332.42 (493.03) 10.51–1902.83 |
| **femoral head** | | | | | |
| DA | 0.58 (0.07) 0.48–0.66 | 0.6 (0.08) 0.51–0.71 | 0.62 (0.1) 0.49–0.75 | 0.65 (0.07) 0.59–0.74 | 0.61 (0.08) 0.48–0.75 |
| Conn.D | 31.38 (18.9) 12.72–67.19 | 23.83 (12.22) 11.06–47.11 | 13.39 (4.76) 8.72–19.92 | 15.98 (4.8) 10.48–23.54 | 22.11 (13.56) 8.72–67.19 |
| BV/TV | 0.4 (0.03) 0.35–0.45 | 0.49 (0.09) 0.32–0.6 | 0.51 (0.02) 0.49–0.54 | 0.46 (0.04) 0.4–0.5 | 0.46 (0.07) 0.32–0.6 |
| Tb.Th.Mean | 0.17 (0.02) 0.13–0.2 | 0.21 (0.03) 0.16–0.27 | 0.24 (0.02) 0.2–0.27 | 0.21 (0.02) 0.17–0.24 | 0.21 (0.04) 0.13–0.27 |
| Tb.Sp.Mean | 0.31 (0.05) 0.25–0.38 | 0.29 (0.08) 0.22–0.44 | 0.33 (0.04) 0.27–0.37 | 0.35 (0.05) 0.32–0.46 | 0.31 (0.06) 0.22–0.46 |
| BS | 922.31 (1207.46) 22.07–3297.5 | 217.52 (371.61) 18.26–1171.76 | 969.85 (1017.6) 153.71–2659.4 | 233.91 (117.24) 114.37–436.48 | 543.21 (820.6) 18.26–3297.5 |
| **humeral/femoral head ratio** | | | | | |
| DA | 0.99 (0.09) 0.88–1.11 | 0.94 (0.23) 0.71–1.48 | 0.86 (0.08) 0.74–0.93 | 0.76 (0.18) 0.46–0.95 | 0.9 (0.18) 0.46–1.48 |
| Conn.D | 1.12 (0.2) 0.9–1.45 | 1.4 (0.5) 0.93–2.35 | 1.32 (0.18) 1.04–1.53 | 1.09 (0.24) 0.79–1.39 | 1.24 (0.35) 0.79–2.35 |
| BV/TV | 0.85 (0.1) 0.72–0.96 | 0.82 (0.08) 0.7–0.95 | 0.81 (0.06) 0.76–0.9 | 0.8 (0.09) 0.7–0.95 | 0.82 (0.08) 0.7–0.96 |
| Tb.Th.Mean | 0.86 (0.1) 0.73–0.98 | 0.8 (0.1) 0.62–0.95 | 0.79 (0.05) 0.72–0.85 | 0.85 (0.04) 0.79–0.92 | 0.83 (0.08) 0.62–0.98 |
| Tb.Sp.Mean | 1.07 (0.09) 0.92–1.17 | 1.1 (0.09) 0.92–1.21 | 1.09 (0.07) 1.01–1.19 | 1.18 (0.2) 0.89–1.47 | 1.11 (0.12) 0.89–1.47 |
| BS | 1.16 (1.04) 0.47–3.46 | 0.53 (0.24) 0.15–0.98 | 0.45 (0.17) 0.23–0.61 | 0.76 (0.39) 0.42–1.39 | 0.73 (0.62) 0.15–3.46 |

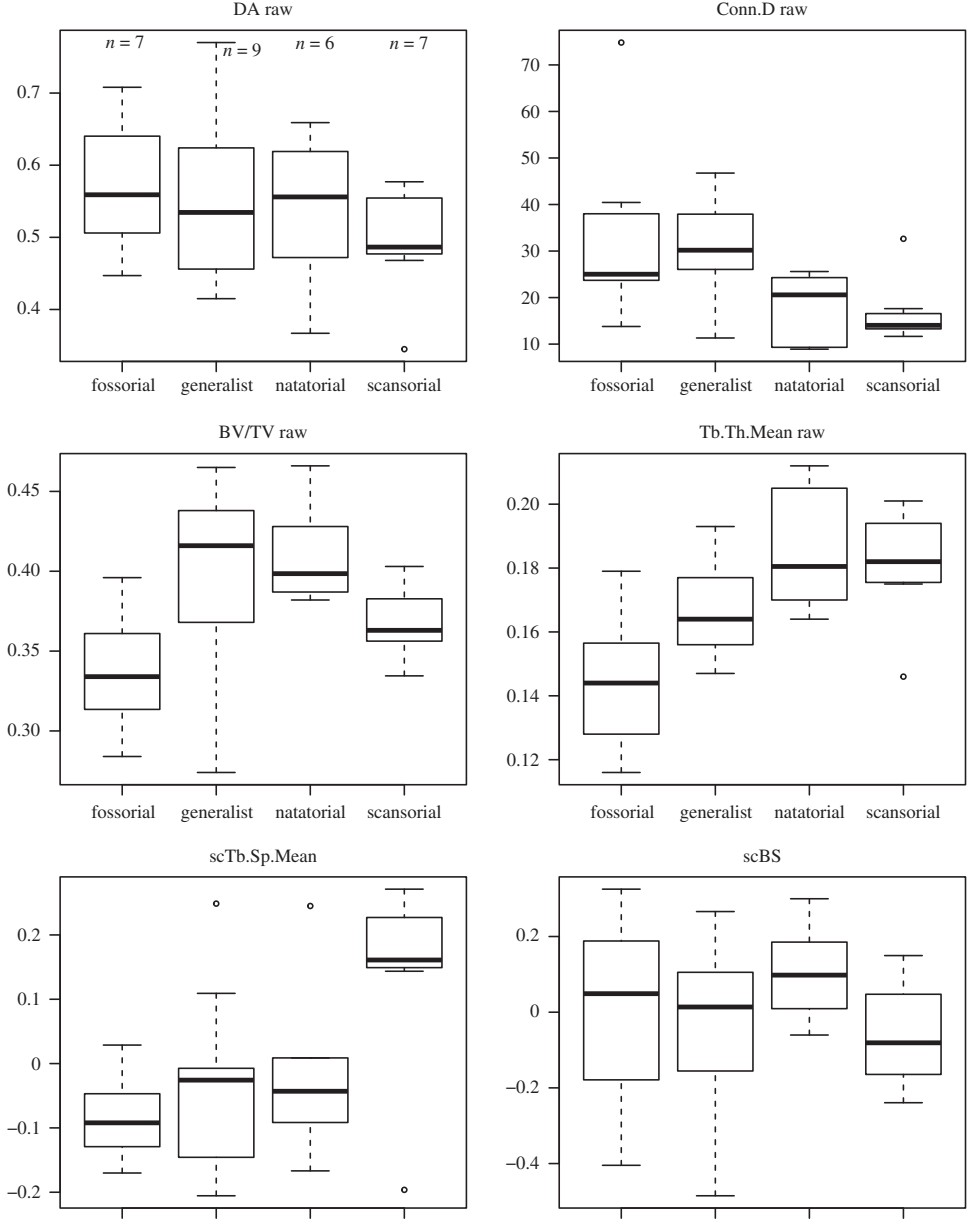

**Figure 3.** Differences of selected trabecular parameters of the humeral head volume of interest (VOI) among mustelid locomotor types. In the case of a significant correlation of the parameter in question with body mass, the residuals of a regression of the parameter against body mass are plotted (name of the parameter is then preceded by the mention 'sc'; these residuals are only used for visualization purposes, see Material and methods). See electronic supplementary material, S8 for other VOIs.

For the femoral head VOI (table 3; electronic supplementary material, S7), there are no significant differences in DA among mustelid locomotor types. With regard to Conn.D, fossors have significantly greater values than other locomotor groups (Fossorial:Generalist, $p$-value = 0.034; Fossorial:Natatorial, $p$-value = 0.017; Fossorial:Scansorial, $p$-value = 0.011). How locomotor groups differ in BV/TV, Tb.Th and Tb.Sp for the femoral head parallels the results for the humeral head VOI (with the difference that all raw values are usually higher for the femoral head), the only difference being that natators' Tb.Th values are significantly higher than those of other locomotor types. Scansors' Tb.Sp values are only significantly greater than those of natators ($p$-value = 0.008). For BS, only the pairwise comparison of fossors and generalists recovers a significant difference ($p$-value = 0.045), with fossors having greater values than generalists.

For the femoral lateral condyle VOI (electronic supplementary material, S8 and S9), DA, Conn.D and BV/TV values do not significantly differ among the locomotor types. Tb.Th values are significantly lower in fossors than in the other types (Fossorial:Generalist, $p$-value = 0.003; Fossorial:Natatorial,

**Table 3.** Result summary of the AN(C)OVA analyses testing for a difference among the locomotor types for each trabecular parameter of interest of the humeral head, femoral head and ratio between these two VOIs (see electronic supplementary material, S9 for all studied VOIs). Abbreviations: SC, whether the parameter is significantly correlated to the estimated specimens' body mass (BMsp, see Material and methods), which also indicates whether an ANOVA (no correlation) or ANCOVA (significant correlation) was run; lambda, Pagel's lambda used to scale the timetree used in the phylogenetic AN(C)OVA (see Material and methods); *p*-value and *F*-value correspond to the influence of the locomotor type on the parameter in question.

| | SC | lambda | *p*-value | *F*-value |
|---|---|---|---|---|
| humeral head | | | | |
| DA | false | <0 | 0.529 | 0.74 |
| Conn.D | false | >1 | 0.002 | 9.33 |
| BV/TV | false | 0.99 | 0.039 | 3.4 |
| Tb.Th.Mean | false | >1 | 0.002 | 7.68 |
| Tb.Sp.Mean | true | 0.81 | 0.027 | 2.57 |
| BS | true | >1 | 0.001 | 0.64 |
| femoral head | | | | |
| DA | false | <0 | 0.392 | 1.02 |
| Conn.D | true | 0.05 | 0.008 | 3.64 |
| BV/TV | false | 0.78 | 0.006 | 5.63 |
| Tb.Th.Mean | false | 0.63 | 0.002 | 7.23 |
| Tb.Sp.Mean | true | 0.77 | 0.004 | 2.84 |
| BS | true | 0.81 | 0.001 | 1.36 |
| humeral/femoral head ratio | | | | |
| DA | false | <0 | 0.06 | 2.65 |
| Conn.D | false | 0.23 | 0.367 | 1.17 |
| BV/TV | false | 0.41 | 0.746 | 0.41 |
| Tb.Th.Mean | false | 0.54 | 0.308 | 1.28 |
| Tb.Sp.Mean | false | <0 | 0.551 | 0.77 |
| BS | false | 0.68 | 0.112 | 2.29 |

*p*-value = 0.034; Fossorial:Scansorial, *p*-value = 0.029). For Tb.Sp, only the pairwise comparison between fossors and scansors yields a significant difference (*p*-value = 0.010), with fossors having lower Tb.Sp values than scansors. As is the case for the humeral head VOI, ANCOVA results indicate that BS should differ among the locomotor types (significant *p*-value of the ANCOVA for that trait); however, pairwise comparisons of the four types uncover no significant pairwise differences.

For the proximal inter-limb ratios (i.e. ratio between the humeral head and femoral head VOIs), only DA differs significantly among the locomotor types. The values gradually decrease in the order of fossors, generalists, natators and scansors (figure 4, table 3), with the only significant difference occurring between fossors and scansors (*p*-value = 0.011). These results indicate that humeral head trabeculae are as anisotropic as those of the femoral head in the fossors, intermediate in generalists and natators and less anisotropic in the scansors. For the distal inter-limb ratios (i.e. ratio between the humeral trochlea and femoral lateral condyle VOIs), the only significant difference was found for BV/TV. Pairwise comparisons of locomotor types find that natators have significantly higher values of BV/TV than generalists and fossors (i.e. humeral trochlea's bone content is relatively high compared to that of the femoral lateral condyle in natators; Generalist:Natatorial, *p*-value = 0.010; Fossorial:Natatorial; *p*-value = 0.029; figure 4; see also electronic supplementary material, S9).

A phylogenetic PCA including parameters from all four studied VOIs, as well as the BMsp, reveals that the regions of trabecular architecture morphospace occupied by each locomotor type overlap quite extensively (figure 5). Interestingly, an inspection of the first three pPCs (representing cumulatively 72.5% of the explained variance) reveals that the generalists occupy a relatively larger region of trabecular morphospace, especially along pPC3. The absolute value of BMsp's loading is higher for pPC1 (0.71; electronic supplementary material, S10 and S11), so one can argue that the other pPCs are less

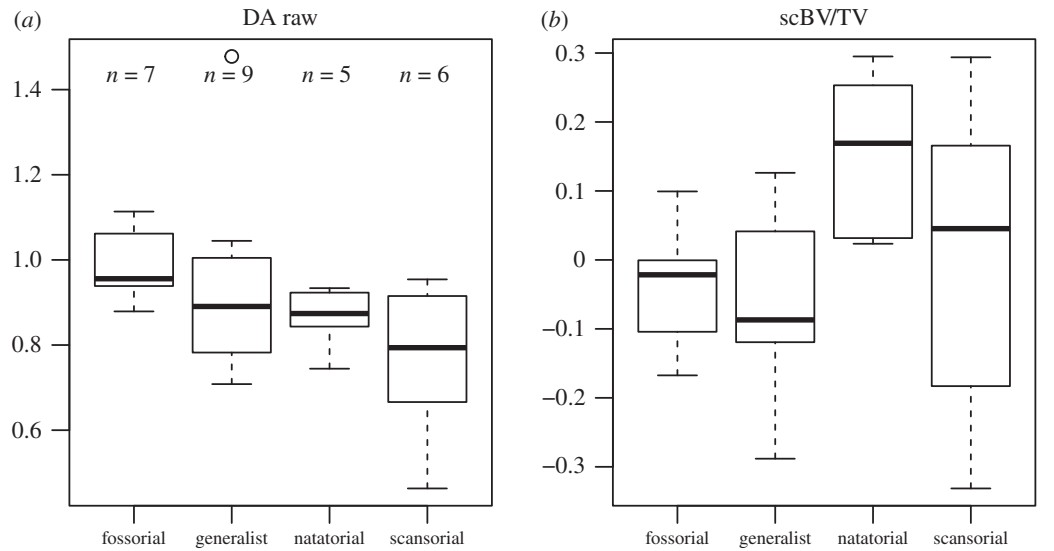

**Figure 4.** Differences of selected trabecular parameters' inter-limb ratios among mustelid locomotor types. (*a*) Humeral head/femoral head ratio of DA. (*b*) Humeral trochlea/femoral lateral condyle ratio of BV/TV. The latter was significantly correlated to body mass, so the residuals of a regression of this parameter against body mass are plotted (these residuals are only used for visualization purposes, see Material and methods).

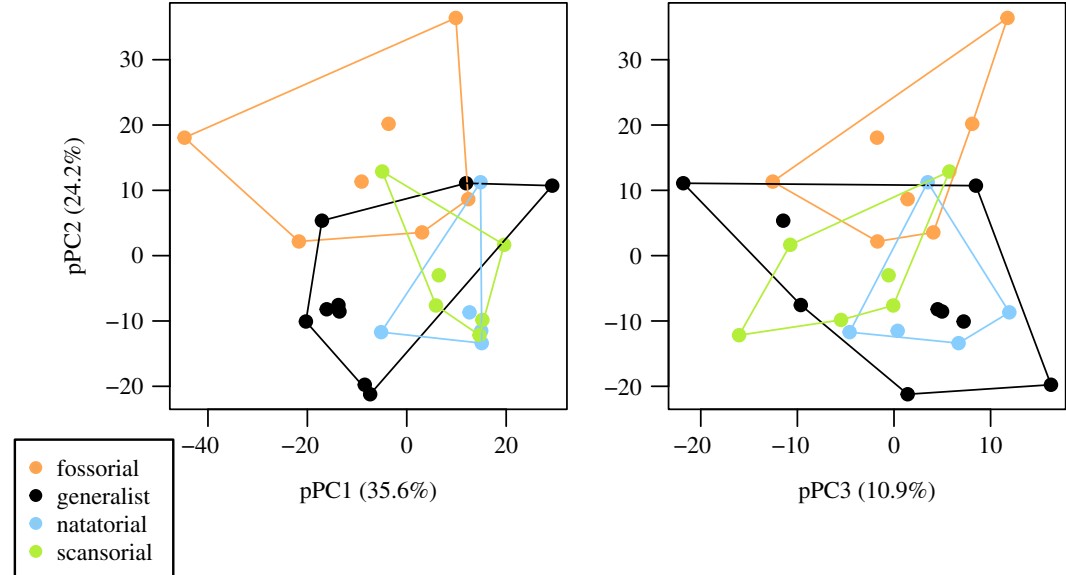

**Figure 5.** Trabecular architecture morphospace as defined by a phylogenetic principal component analysis. Trabecular parameters from all four studied VOIs as well as the BMsp (specimens' estimated body mass) were used for this analysis. The two plots present the three first phylogenetic principal components (pPCs).

affected by size. Some types can be distinguished along these other pPC axes, with the natatorial, scansorial and fossorial taxa separate from one another to some extent along pPC2 or pPC3. Fossorials are moreover partly discriminated from the other locomotion types along pPC2.

# 4. Discussion

## 4.1. Confrontation to expectations

With this analysis, we aimed at unravelling the potential relationships between trabecular bone architecture and locomotor types within a diverse group of closely related mammals, the mustelids.

**Table 4.** Ranges of degree of anisotropy (DA) and bone volume fraction (BV/TV) in other mammalian clades.

| | | DA | BV/TV |
|---|---|---|---|
| Mustelidae[a] | humeral head | 0.34–0.77 | 0.27–0.47 |
| | femoral head | 0.48–0.75 | 0.32–0.6 |
| Primates[b] | humeral head | 0.06–0.53 | 0.12–0.48 |
| | femoral head | 0.19–0.81 | 0.20–0.77 |
| Xenarthrans[c] | humeral head | 0.24–0.79 | 0.29–0.60 |
| Sciuromorphs[d] | femoral head | 0.45–0.88 | 0.20–0.61 |

Data references: [a]this study.
[b][31,40,41,46,55–57].
[c][25].
[d][26].

A rather poor discrimination of our *a priori* locomotion-based groupings (i.e. generalists, fossors, natators and scansorials) was recovered (figure 5). This could be explained, at least in part, by the fact that the range of variation described by some of the acquired trabecular parameters, the DA or the bone fraction (BV/TV), for instance, is relatively restricted when compared to other clades for which trabecular parameters were similarly acquired (see below and table 4; note that for some of these other clades, parameter acquisition followed a different methodology). Nevertheless, some interesting differences among these groups for some of the trabecular parameters were recovered and will be here discussed.

Contrary to our expectations, DA was not found to be consistently greater in fossorial mustelids' forelimb. The only significant difference in DA concerns the humeral trochlea and proximal inter-limb ratios. For the latter, fossors are indeed the only sampled mustelids for which DA is roughly as high in the humeral head as in the femoral head (figure 4, table 2), with the exception of one outlying generalist (*Mustela eversmanii*). Interestingly, a greater DA was associated with fossoriality in xenarthrans (only the forelimb was investigated in their case; [25]). Therefore, even though there is not a consistently higher DA in fossorial mustelids, the relatively high values yielded by their humeral head might indicate, as in xenarthrans, that strenuous digging activity is to some (lower) extent transcribed in their trabecular architecture. Beside this inter-limb ratio difference for DA, only one other of such ratios was found to differ among locomotor types: BV/TV of the distal inter-limb ratio. The lack of locomotion-related differences in most inter-limb ratios investigated cannot be explained by a simple covariation between the limbs (as was demonstrated regarding their morphology; [10]), because the obtained ratios often deviate from 1 (table 2, electronic supplementary material, S7). This is true, in particular, for the proximal inter-limb ratio of DA, which indicates that humeral head DA can be either greater or lower than that of the femoral head. That differs from the pattern observed in anthropoids, for which lower values of DA in the humeral head are consistently recovered [41,55]. This was interpreted in the latter clade as possibly indicative of their mobile shoulder joint and manipulative forelimb. Our results seem to be compatible with such an interpretation, but sampling other carnivorans with better manipulative abilities [15] would be needed to confirm it. This is also consistent with the fact that DA does not seem to be subject to systemic differences [58]. One taxon with notable manipulative abilities sampled here is the sea otter (*Enhydra lutris*; [59]). As the ratio between its humeral head and femoral head DA values are not particularly high (similar to that of most weasels for instance), this taxon does not seem to follow the pattern described for anthropoids.

BV/TV variation among mustelids partly follows our expectations, with greater values found in the natators for some of the VOIs (humeral head and femoral head). But generalists were also usually characterized by high values. And, unexpectedly, natators also differed from other locomotor types in featuring humeral trochlea containing more bone than the femoral condyle (figure 4, table 2). The sea otter's values were not exceptionally high (0.39 and 0.51 for the humeral and femoral heads, respectively) compared to other natators (0.41 and 0.51 being the respective mean values for the humeral and femoral heads). This may relate to the sea otter relying on its hindlimbs to swim [60], though it should be noted that the forelimbs are used in manipulating and tool use [61], which may nonetheless subject the forelimb skeleton to large forces. A trend of increasing long bone compactness was documented in aquatic mustelids when compared to more terrestrial members of the family, this

trend potentially stemming from the influence of body size [62]. There was no effect of size on BV/TV for all but one VOIs we investigated (table 3; electronic supplementary material, S9), so the increased bone content in limb bones related to aquatic habits seems to be confirmed by our analysis.

These results are in line with those of Kilbourne & Hutchinson [8], who documented a greater dimensionless cross-sectional area for the forelimb bones in natatorial mustelids. This would suggest that this increase of bone content affects both cortical and trabecular bone of limb long bones. A similar pattern was also described for the overall diaphyseal microanatomy of aquatic mustelids' humerus and femur, with a special emphasis on the sea otter ([19]; therein the equivalent of bone fraction are the three-dimensional compactness and whole compactness parameters) and in the humerus of carnivorans (including mustelids; [63]). Regarding long bone gross morphology, natatorial and fossorials species are viewed as being characterized by a robust forelimb (the humerus in particular is short and robust; [7,17]; locomotor types termed 'aquatic' and 'semi-fossorial' therein). In the case of otters, this gross morphological feature was emphasized in the context of a potential ballast [18]. Bone mass increase was also documented in the sea otter's ribs (overall compactness = 0.91, versus 0.65–0.72 for *Martes martes*, *Martes foina*, *Neovison vison*; [64]), but not in the mid-lumbar vertebra (termed therein 'relative area of mineralized bone'; [65]). This could suggest that bone mass increase is restricted to some regions of the skeleton in this highly specialized aquatic taxon. Nevertheless, our phylogenetic PCA (figure 5) does reveal that the sea otter departs from other natators due to a more positive value along pPC2. This pPC is inversely correlated to bone fraction parameters, in particular, those of the humeral head and femoral head VOIs (electronic supplementary material, table S10). Inspection of the trabecular parameters among otters shows that BV/TV values are lower in the sea otter than in the other natatorial mustelids for the distal VOIs we have investigated. The osteosclerosis described for ribs and limb bones' diaphysis in the sea otter should therefore be regarded as not affecting the bone fraction of the trabecular architecture in the humeral and femoral epiphyses. While not about epiphyseal but metaphyseal trabeculae, it appears that greater volume fraction also characterizes some of the more aquatic reptiles [66]. Indeed, crocodylians and (semi-)aquatic squamates and turtles were usually found as exhibiting higher values for this parameter than other sampled reptiles. Furthermore, bottom-walking turtles were identified as characterized by particularly high values. This could indicate that bone mass increase, an adaptation well documented in long bone diaphyses of some (semi-)aquatic amniotes, can more generally concern the overall structure of long bones. It is noteworthy that bone mass increase can also affect parts of the skeleton of semi-aquatic amniotes outside of the limbs, such as the cranial bones [67].

Variation in BV/TV does not seem to describe systemic patterns in mustelids, as found in chimpanzees and humans [58]. But, as is the case in anthropoids [41], mustelid BV/TV and Tb.Th values are consistently found to be lower in the humeral head than in the femoral one. This could indicate that compressive forces are higher at the hip joint than at the shoulder joint, as in primates [41]. Given the pattern observed in other terrestrial mammals [68], greater compressive forces at the hip joint are not expected for mustelids. However, to our knowledge, these forces have not been quantified in mustelids, so new experimental data are required to better understand the distribution of bone fraction between the fore- and hindlimb in relation to locomotion. But the peak vertical forces measured during the domestic ferret's (*Mustela putorius*) epigean and subterranean locomotion [69] nevertheless suggest that mustelids do not differ in that regard from other terrestrial non-primate mammals.

An interesting outcome of our analysis is that generalist mustelids occupy a larger region of trabecular morphospace (as shown by phyl.PCA; figure 5). This pattern was not found in a three-dimensional GM analysis of the musteloid forelimb [17] (therein called 'Terrestrial'), but it is reminiscent of the results based on mustelid forelimb linear measurements [7]. Regarding trabecular architecture, the various activities performed by generalists and their lack of specialization for a particular function (e.g. climbing, digging or swimming) may possibly preclude a restricted range of trabecular parameters values that can be associated with a more specific specialization. Sampling several individuals within each of the species that are generalist (e.g. *Mustela* sp., *Galictis* sp., *Vormela peregusna* and *Gulo gulo*) is necessary to determine whether each species regarded as generalist is characterized by a distinct trabecular architecture within this broader region of morphospace or if generalists show greater intraspecific variation. Our sampling was designed to encompass the taxonomic diversity of mustelid. Because 22 out of the 29 species are represented each by only one specimen, understanding of the intraspecific variation is very limited. Though our study focused on the macroevolution of trabecular parameters, intraspecific variation in trabecular parameters is a topic worthy of future study in its own right.

## 4.2. Comparison to other clades

In order to compare how the trabecular architecture of the mustelids sampled here relates to that of other clades, we gathered the acquired data previously published for the latter (table 4). One should note that this usually entails different VOI definition across the analyses, and the following should therefore be viewed as rough comparisons. The mean humeral head DA among mustelids is fairly similar to that of xenarthrans and hence greater than that commonly featured by primates [25]. However, BV/TV is lower for this VOI in mustelids than in xenarthrans, falling in the range of primates ([31,40,41,46,55–57]; table 4) which might emphasize the particularly high bone content already signalled for xenarthrans in general [70–73].

The mean femoral head DA and BV/TV of mustelids are similar to those of sciuromorphs [26]. In primates, the femoral head DA and BV/TV have shown a quite wide range of variation (table 4), which encompasses that of mustelids. The relatively restricted range of values for these parameters described by mustelids and sciuromorphs might hence be related to fully quadrupedal locomotion; however, this hypothesis should be tested with a dataset including bipedal animals.

As in anthropoid primates [41], BV/TV is consistently found as higher in the femoral head than in the humeral one. However, differing from primates, the humeral head trabecular architecture can be roughly as isotropic as that of the femoral head in mustelids, especially in fossorial taxa.

## 4.3. Evolutionary patterns

The evolutionary patterns described by mustelid trabecular parameters can be drawn from phylogenetic character mapping (electronic supplementary material, S6). The humeral head of fossors and generalists is characterized by greater values of Conn.D. For fossors, this seems to be mostly due to a condition independently acquired in the American badger (*Taxidea taxus*) and the sampled Ictonychinae. Sampling non-mustelid musteloids would allow for assessing whether this represents a convergence or a reversal (in ictonychines). The overall pattern could alternatively be explained by a convergent trabecular diminution in otters and the Guloninae, but the fact that other basal fossorial mustelids (*Arctonyx*, *Meles*, *Mellivora*) show fairly low values seems to contradict such a hypothesis. Furthermore, the femoral head Conn.D values are also found as particularly high in the non-*Galictis* ictonychines, which can indicate that at least this clade is marked by a derived condition regarding this trabecular parameter. The fact that greater values of BV/TV are found for natators and generalists' humeral head and femoral head is actually driven by a convergence between some weasels, particularly those of the genus *Mustela*, and some otters. High values of Tb.Sp, characterizing the scansorials' humeral head and femoral head, are also found in the wolverine (*Gulo gulo*), which could indicate that the condition is plesiomorphic for the Guloninae (the greater grison, *Galictis vittata*, also features particularly high values). Values of Tb.Sp are also found as particularly low for fossors' femoral lateral condyle. This pattern seems to be mainly due to a convergent decrease of this parameter in the honey badger (*Mellivora capensis*), the Chinese ferret-badger (*Melogale moschata*) and the striped polecats (*Ictonyx* sp.). Relatively high Tb.Th values are found for the latter VOI in scansors, but this is also true for the wolverine. This suggests that such a trait might be ancestral for the subfamily (Guloninae).

The European mink (*Mustela lutreola*) is the only non-otter natator included in our dataset. Its trabecular parameters were generally found as more similar to those of the other members of the genus (classified *a priori* as generalist) than those of otters. This could be due to the fact that European mink is viewed as less specialized for swimming than otters, which could question its assignation in the same locomotor type. It should also be noted that when sympatric with Eurasian otters (*Lutra lutra*), N. American mink (*Neovison vison*) tend to shift from aquatic foraging and a more piscivorous diet to more terrestrial foraging and a more generalist diet [74]. Thus, a more unspecialized limb skeleton may possibly be tied to the shifts in a dietary niche in mink species.

## 4.4. Body elongation

It was recently emphasized that body elongation ratio was an important aspect of musteloid evolution [12]. Furthermore, forelimb length was found as reduced in correlation with body elongation (a negative relationship between forelimb length and body length). This can be seen as well in our dataset, where the total volume of the humeral head VOI (mean = 71.4 mm$^3$) for most species is lower than the corresponding femoral head VOI (mean = 111.3 mm$^3$). The VOIs were selected to be as large as possible and centred around a point defined as the centre of the articular surface and are hence

expected to scale proportionally to the size of the articular structure (see proximal inter-limb ratio, electronic supplementary material, S1). Inspection of the mean Tb.Th values reveals that a reduction of the forelimb seems to entail thinner trabeculae, because this parameter's values are lower for the humeral than the femoral head for all specimens but one (for which the ratio is very close to 1). Correspondingly, Th.Sp and Conn.D values are higher for the humeral than the femoral head for most specimens (85% and 71%, respectively). However, Tb.Th values for the proximal inter-limb ratio in otters are not particularly low (neither Tb.Sp nor Conn.D values are particularly high), which could have been expected given the stronger reduction of their forelimb when compared to other mustelids [12].

# 5. Conclusion

Our analysis of the humeral and femoral epiphyses of mustelids reveals that a clear discrimination of locomotor types found in this clade solely based on trabecular bone parameters is difficult. Nevertheless, we documented a relationship between some trabecular bone features and some of these locomotor types. The bone fraction, in particular, seems to be especially high in natators. This could suggest that bone mass increase, documented as an aquatic adaptation for the long bones' diaphysis of some amniote clades, might also affect epiphyseal trabecular bone. Adding extinct mustelids to the sampling presented herein could allow for a better understanding of the phenotypic evolution of the clade, moreover, particularly allowing us infer whether the gross morphological and trabecular adaptations associated with lifestyle transitions are acquired jointly or independently, and in the latter case, the chronology of acquisition of these traits for adaptations.

Data accessibility. All measurements and code generated for this study have been uploaded as part of the electronic supplementary material. Raw CT scans are available on Morphosource (https://www.morphosource.org/Detail/ProjectDetail/Show/project_id/674). Scans performed at the Museum für Naturkunde Berlin ( table 1) are also curated in the public digital collection of this institution (contact person: Kristin Mahlow; kristin.mahlow@mfn-berlin.de).

Authors' contributions. B.M.K. and E.A. designed the study, drafted the manuscript, critically revised the manuscript, gave final approval for publication and agree to be held accountable for the work performed therein. B.M.K. acquired CT scans and prepared them for further analysis. E.A. supervised and took part in parameter acquisition and ran the analyses.

Competing interests. The authors have no competing interest to declare.

Funding. This study was funded by the Deutsche Forschungsgemeinschaft (grant nos. AM 517/1-1, KI 1843/3-1 and KI 1843/3-2) and the Museum für Naturkunde Berlin's Innovation Fund. The publication of this article was funded by the Open Access Fund of the Leibniz Association.

Acknowledgements. For access to specimens, we thank Steffen Bock, Christiane Funk, Frieder Mayer and Detlef Willborn, (MfN Berlin); Rebecca Banasiak, Adam Ferguson, Larry Heaney, Bruce Patterson and John Phelps (Field Museum of Natural History); Daniel Klingberg Johansson and Eline Lorenzen (Natural History Museum Denmark); and Petr Benda (Národní Muzeum). For assistance with CT scanning, we thank Martin Kirchner, Kristin Mahlow and Johannes Müller (MfN Berlin) and Zhe-Xi Luo and April Neander (University of Chicago). We thank Jenny Michel and John Nyakatura for helping in data acquisition. We also thank Mike Collyer for his help with using the rrpp package and John Nyakatura, Léon Botton-Divert and Jan Wölfer for comments and discussion.

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
