## [Reviewer comments · Royal Society Open Science]

Review History

RSOS-190938.R0 (Original submission)

Review form: Reviewer 1

Is the manuscript scientifically sound in its present form?

Yes

Are the interpretations and conclusions justified by the results?

No

Is the language acceptable?

Yes

Is it clear how to access all supporting data?

Yes

Do you have any ethical concerns with this paper?

No

Have you any concerns about statistical analyses in this paper?

I do not feel qualified to assess the statistics

Recommendation?

Major revision is needed (please make suggestions in comments)

Comments to the Author(s)

This paper quantifies the trabecular structure within the proximal and distal epiphyses of the humerus and femur in mustelids. They sample 36 bones across a broad sample of 29 species, which they divide into four locomotor groups: natatorial, fossorial, scansorial and generalists. They look at the relationship between trabecular parameters and variation in body mass, and how trabecular parameters vary across the mustelid phylogeny. They find limited evidence for differences in trabeculae across the different locomotor groups, apart from higher BV/TV in natatorial species.

Overall, this is an interesting study and provide novel data on an interesting group mammals that vary substantially in the locomotor behaviours. The authors provide a good broader context on how mustelid trabecular morphology compares to other mammals. I like that the authors have considered variation in body mass and the phylogenetic relationships among the study taxa when analysing/interpreting their data. The writing within the Introduction especially, and generally the Discussion is clear and concise. I have, however, a few major concerns about hypotheses and methods and there were several points where it was unclear to me what was being done or why. I have outlined these, as well as more minor comments, below. Mustelids are not my field of expertise, which likely added some additional confusion that would not be expected from an expert. However, given the broad readership of this journal, it is worth defining terminology or explaining predictions (e.g. greater bone for buoyancy) that non-experts may not be familiar with. First and foremost, I would recommend defining “natatorial”, “fossorial”, and “scansorial”. Also, there are a LOT of abbreviations in this manuscript. Removing some of them will make it much more reader-friendly.

I find the study predictions (pg 4) difficult to follow. It is not clear to me why natatorial mustelids are predicted to have greater BV/TV, TbTh and Conn.D, and lower Tb.Sp when, as aquatic tetrapods, buoyancy is considered to be the most important functional constraint. I would think less bone, rather than more, might be better for buoyancy? (see comment above). The prediction about natatorial mustelids seems to be the same as that of fossorial taxa, which are also predicted to have greater BV/TV, TbTh, Conn.D and lower Tb.Sp (lines 19-20). How do you expect that the fossorial and natatorial taxa will compare/differ?

When one gets to the Methods, it is clear that bone surface area (BS) and mean direction of the trabeculae (MDT) are also quantified, but they are not included in the predictions. Why not? Are these variables adding any useful information/results that are not already quantified in the other parameters? Maybe so, but just worth considering...just because they are quantified in BoneJ doesn't mean that they all need to be included. In particular, predictions about MDT could be made based on difference in joint angles or joint load perhaps? Currently, MDT is reported in the results as showing no clear differences across the locomotor groups, but without any context as to whether this is an expected or surprising results. Either link MDT to something specific about the locomotor groups, or leave it out.

The Methods describe “This macro was used to extract a substack centred in each epiphysis that is as big as possible without including cortical bone. The epiphyses of some small-sized mustelids only contained a few trabeculae. We therefore chose this approach to acquire the ‘bulk’ of the epiphyseal trabeculae.” I understand the constraints of size and morphological variation across different taxa and thus why the authors have chosen this method. However, I would like to see

some intra/interspecific assessment of how well this method works and how well it avoids the over-sampling problem (see Fajardo and Muller, 2001 *Am J Phys Anthropology*). For example, were intra/inter-observer error tests conducted to see if the ROI could be placed in the same position over multiple trials or how much the trabecular parameters varied with slight differences in ROI position? Furthermore, the ROI in the proximal humerus and, less so, the femoral condyle, is very small compared to the size of the epiphysis/amount of trabeculae that is NOT quantified so I don't think this can be described as a "bulk" approach. ROI size is particularly important when one gets to the Discussion, with statements like "The ROIs were selected to comprise the bulk of the epiphysis' trabeculae, and are hence expected to be scale proportionally to the size of the epiphysis (see proximal interlimb ratio, SOM 1).", in which the authors are specifically referring to the humeral and femoral heads. The relative amount of trabeculae being quantified in each is very different, according to Figure 2.

Figure 2 is very useful, but it would be helpful to see an example of the ROI placement in one of the smallest taxa and one of the largest, to see how exactly how homologous the ROI placement and size is from a morphological perspective. I think such tests/figures are particularly important since only one specimen represents the trabecular parameters for most of the study species.

In the results section, I am a bit confused by the first sentence (pg. 8): "The lifestyles represented by the mustelids we sampled indicate that the ancestral condition for the clade was ostensibly fossorial, and that five transitions - to the generalist, natatorial, and scansorial specializations, and one reversal to fossoriality - have occurred during their evolutionary history (Fig. 1)." Are the phylogenetic relationships among mustelids not known from genetic data already? Mustelids are outside my research field, so I do not know, but the first sentence suggests that the phylogeny is being created using the locomotor specialisations first, and then mapping the trabecular parameters onto that phylogeny, rather than onto the "true" phylogeny. Is that correct? Or are the locomotor specialisations mapped onto the existing "true" phylogeny, which is what seems to be depicted in Figure 1 (but it's not clear). If it's the latter, has this not been done before? I'm just wondering if the "five transitions" found here is also novel information, or if previous research has determined the evolutionary locomotor history across mustelids and that therefore, this is information that can be presented in the Introduction. Also, the phylogenetic analysis is described at the very end of the Methods section, but presented at the beginning of the Results section. Since the focus on the paper is on the trabecular structure, it makes sense (to me anyway) to describe the trabecular structure first?

The Results section needs re-working. It is challenging to follow, the language at times is a bit colloquial/grammatically incorrect, and lacks necessary detail in some places. Eg. pg. 9, line 26 "... with differences that natators' Tb.Th values are even higher than those of other types". Is this a significant difference? Are natatorials significantly higher than ALL the other locomotor groups, or just some of them? Same at line 29: "BS values are only found as higher in fossors than in generalists". Is this significant? And should be written as "BS values are only significantly higher in fossors compared with generalists." I think p values could be included within the text throughout where significant differences are found.

It is unclear to me in the Results section what the body mass results actually are and how they relate to the trabecular parameters specifically. More information in the legend of Figure 5 is needed to tell the reader exactly how this PCA tells us about body mass effects. Related to this, the Discussion states "There was no effect of size on BV.TV for all but one ROIs we investigated (Table 2),..." (pg 12, line 38) but I do not see body mass data in Table 2.

The Discussion starts with a comparison to other clades (e.g. primates and zenarthrans) before actually discussing the results of mustelids. It would be better to present the results of this study first, and then move to the broader comparative context. Importantly, the Discussion needs to

include some acknowledgment of the fact that 22 of your 29 species are represented by $n=1$. This paper balances taxonomic breadth of the sample with small size, so it's understandable, but one needs to clearly acknowledge the limitations of such a small sample.

Additional comments:

The introduction does not include any information about what "long bones" are being studied. This should be clarified in the Introduction, since there are many more long bones in the skeleton than just the humerus and femur.

Throughout the paper, since this is a three-dimensional analysis of trabecular bone, "region of interest" (ROI), would be more accurately called a "volume of interest" (VOI).

Throughout the paper the "locomotor groups" are referred to variously as "specialisations" or "types" or "locomotor specialisations" or "groupings". Please pick one and use it consistently.

pg 3, ln 51: for this general, fundamental statement it would be good to cite some of the more classic literature/researchers on this topic, such papers/books by Currey, Rubin, Lanyon, Frost, etc. Currently only one experimental study by Barak et al. (2011) is cited.

pg. 4, ln 3: "Tb.Sp" stands for "trabecular separation", not "trabecular spacing". The standard abbreviation is "BV/TV", not "BV.TV", since it's a ratio.

pg. 4, ln 31: "greater" than what? Perhaps change to simply "resist HIGH mechanical loads"

pg. 5, line 10: change "good enough" to "sufficient"

pg. 5, line 15: The statement "only right bones were used" is not true if you also used left bones and mirrored them. Please re-write, e.g. "Right bones were used when possible and, if not, the left bone was used and mirrored".

pg. 5, line 17: Please clarify what is meant by "measurements". What kind of measurements?

Review form: Reviewer 2 (Alexandra Houssaye)

Is the manuscript scientifically sound in its present form?

No

Are the interpretations and conclusions justified by the results?

Yes

Is the language acceptable?

Yes

Is it clear how to access all supporting data?

Yes

Do you have any ethical concerns with this paper?

No

Have you any concerns about statistical analyses in this paper?

Yes

Recommendation?

Major revision is needed (please make suggestions in comments)

Comments to the Author(s)

Dear Editor,

This study proposes to analyze the trabecular organization in the epiphyses of stylopod long bones in mustelids and notably focuses on the functional interpretations of the results. The study is carefully carried out and well presented. However, I have various points of concern.

I notably do not understand their body mass estimation. By making a regression between the mass taken from the literature and one parameter of their bone, they create a “rule” which suggests that the same relationships exists between this parameter and mass in all their taxa; the regression is phylogenetically informed, which can reduce this “overall rule”, correcting it by the phylogeny. However, to my knowledge none of the numerous body mass estimation formulae existing (from long bones [length, circumference.....], astragalus, teeth...) takes the phylogeny into account because proportions between a single skeletal element and mass have not so great chances to follow the phylogeny (although, for some of these formulae, some are restricted to some groups [e.g. artiodactyles, primates, carnivorans], which remains different than considering an impact of phylogeny). The approach sounds rather “heavy” in assumption for me. Moreover, if they have for one species a small sample, for another a giant one... it will add biases. I do not see why they do not rather estimate body mass using one of the numerous formulae; considering they have the stylopod long bones they can easily estimate it based on their two circumferences. In addition, here the parameter taken seems to be TV, which represents the total volume of the ROI, and which is thus subjective, not homologous between bones... BS would appear more easily “homologous”. I thus strongly recommend to estimate mass differently.

I also do not understand the way the authors deal with size for some parameters. If a correlation between size and the absolute thickness of the trabeculae naturally exists, I do not understand how they deal with this size effect. They do a linear regression and work on the residuals. This is often done for geometric morphometrics data when one variable does not necessarily correspond truly to one anatomical feature. However, here again, they thus consider a linear relationships between size and this parameter in all their sample. The “rule” is sample dependent of course. The correlation can vary between taxa. I do not understand why they cannot for example transform this parameter (in mm) influenced by size by making it a ratio; e.g. by dividing it by a size parameter (as homologous parameter as possible for size). This does not impose a similar relationship between size and this parameter (Tb.Th) but usually corrects the size effect, though some allometry can remain. If the authors prefer their method, could they justify why.

I also do not understand the inter-limb rations. If the authors want to compare the structure of the humeral and femoral heads, dividing all (raw) parameters of one ROI by those of the other ROI appears very weird to me. Moreover, are the ROI of similar size?? The bones are not. This can have consequences. They could maybe do analyses of covariation like RV tests.

A more minor point, the authors say that collinear variables are excluded from the PCA; please justify this choice; and, especially, please say which ones (I do not recall covariation parameters in annex but maybe they are available). Also, for the PCA visualization, it would be great to add the variables on the graphs, and thus to see the contribution of each variable.

About the material... determining categories is awful, because of course there are continuums. But it is necessary. However, *Mustela lutreola*, though semi-aquatic, appears adapted to a much lesser extent to an aquatic locomotion than otters. I think it might bias results considering the aquatic group. Did you try without this taxon in this category; it is probably much more different from other categories. Because the anatomy of *M. lutreola* is poorly distinct from that of other mustelines.

In the discussion, it would be great to add details about the way you compare the ranges of variation between taxa in your study and data from other studies on other animals/bones. Moreover, the ROIs differ between all these analyses. Please be more specific in the comparisons. Also, please clarify between primates and anthropoids so that the reader really understands which type of primates you are talking about in each case.

I also added a few minor comments directly in the pdf. Please indicate the reference for the phylogeny on figure 1.

After consideration of these comments and a few changes, this study will be a nice contribution, adding data concerning mustelids to a for now still small data basis about the 3D trabecular organization in limb long bones of various amniotes.

I remain at your disposal if you have any comment or question.

Best wishes,

Alexandra Houssaye

Decision letter (RSOS-190938.R0)

24-Jun-2019

Dear Dr Amson,

The editors assigned to your paper ("Trabecular bone architecture in the stylopod epiphyses of mustelids (Mammalia, Carnivora)") have now received comments from reviewers. We would like you to revise your paper in accordance with the referee and Associate Editor suggestions which can be found below (not including confidential reports to the Editor). Please note this decision does not guarantee eventual acceptance.

Please submit a copy of your revised paper before 17-Jul-2019. Please note that the revision deadline will expire at 00.00am on this date. If we do not hear from you within this time then it will be assumed that the paper has been withdrawn. In exceptional circumstances, extensions may be possible if agreed with the Editorial Office in advance. We do not allow multiple rounds of revision so we urge you to make every effort to fully address all of the comments at this stage. If deemed necessary by the Editors, your manuscript will be sent back to one or more of the original reviewers for assessment. If the original reviewers are not available, we may invite new reviewers.

- Data accessibility

If you wish to submit your supporting data or code to Dryad (<http://datadryad.org/>), or modify your current submission to dryad, please use the following link:
<http://datadryad.org/submit?journalID=RSOS&manu=RSOS-190938>

- Competing interests

- Authors' contributions

- Acknowledgements

- Funding statement

on behalf of Dr Michael Doube (Associate Editor) and Kevin Padian (Subject Editor)
 openscience@royalsociety.org

Associate Editor's comments (Dr Michael Doube):

Dear Dr Amson,

Two reviewers have given detailed recommendations and made several well-founded criticisms of this manuscript. I agree with the concerns around estimating body mass from a bone parameter, when the relation between body mass and the bone parameter is likely to vary by phylogeny, and potentially with the lifestyle categories you have selected. It would be worthwhile identifying where your interpretations rely on assumptions or error-prone interpolation and move towards using direct measurements rather than derived terms for your analysis.

Please revise the manuscript, and in your response list a reply to all of the reviewers' comments.

Kind regards,
 Michael Doube

Comments to Author:

Reviewers' Comments to Author:
 Reviewer: 1

Comments to the Author(s)

This paper quantifies the trabecular structure within the proximal and distal epiphyses of the humerus and femur in mustelids. They sample 36 bones across a broad sample of 29 species, which they divide into four locomotor groups: natatorial, fossorial, scansorial and generalists. They look at the relationship between trabecular parameters and variation in body mass, and how trabecular parameters vary across the mustelid phylogeny. They find limited evidence for differences in trabeculae across the different locomotor groups, apart from higher BV/TV in natatorial species.

Overall, this is an interesting study and provide novel data on an interesting group mammals that vary substantially in the locomotor behaviours. The authors provide a good broader context on how mustelid trabecular morphology compares to other mammals. I like that the authors have considered variation in body mass and the phylogenetic relationships among the study taxa when analysing/interpreting their data. The writing within the Introduction especially, and generally the Discussion is clear and concise. I have, however, a few major concerns about hypotheses and methods and there were several points where it was unclear to me what was

being done or why. I have outlined these, as well as more minor comments, below. Mustelids are not my field of expertise, which likely added some additional confusion that would not be expected from an expert. However, given the broad readership of this journal, it is worth defining terminology or explaining predictions (e.g. greater bone for buoyancy) that non-experts may not be familiar with. First and foremost, I would recommend defining “natatorial”, “fossorial”, and “scansorial”. Also, there are a LOT of abbreviations in this manuscript. Removing some of them will make it much more reader-friendly.

I find the study predictions (pg 4) difficult to follow. It is not clear to me why natatorial mustelids are predicted to have greater BV/TV, TbTh and Conn.D, and lower Tb.Sp when, as aquatic tetrapods, buoyancy is considered to be the most important functional constraint. I would think less bone, rather than more, might be better for buoyancy? (see comment above). The prediction about natatorial mustelids seems to be the same as that of fossorial taxa, which are also predicted to have greater BV/TV, TbTh, Conn.D and lower Tb.Sp (lines 19-20). How do you expect that the fossorial and natatorial taxa will compare/differ?

When one gets to the Methods, it is clear that bone surface area (BS) and mean direction of the trabeculae (MDT) are also quantified, but they are not included in the predictions. Why not? Are these variables adding any useful information/results that are not already quantified in the other parameters? Maybe so, but just worth considering... just because they are quantified in BoneJ doesn't mean that they all need to be included. In particular, predictions about MDT could be made based on difference in joint angles or joint load perhaps? Currently, MDT is reported in the results as showing no clear differences across the locomotor groups, but without any context as to whether this is an expected or surprising results. Either link MDT to something specific about the locomotor groups, or leave it out.

The Methods describe “This macro was used to extract a substack centred in each epiphysis that is as big as possible without including cortical bone. The epiphyses of some small-sized mustelids only contained a few trabeculae. We therefore chose this approach to acquire the ‘bulk’ of the epiphyseal trabeculae.” I understand the constraints of size and morphological variation across different taxa and thus why the authors have chosen this method. However, I would like to see some intra/interspecific assessment of how well this method works and how well it avoids the over-sampling problem (see Fajardo and Muller, 2001 Am J Phys Anthropology). For example, were intra/inter-observer error tests conducted to see if the ROI could be placed in the same position over multiple trials or how much the trabecular parameters varied with slight differences in ROI position? Furthermore, the ROI in the proximal humerus and, less so, the femoral condyle, is very small compared to the size of the epiphysis/amount of trabeculae that is NOT quantified so I don't think this can be described as a “bulk” approach. ROI size is particularly important when one gets to the Discussion, with statements like “The ROIs were selected to comprise the bulk of the epiphysis’ trabeculae, and are hence expected to be scale proportionally to the size of the epiphysis (see proximal interlimb ratio, SOM 1).”, in which the authors are specifically referring to the humeral and femoral heads. The relative amount of trabeculae being quantified in each is very different, according to Figure 2.

Figure 2 is very useful, but it would be helpful to see an example of the ROI placement in one of the smallest taxa and one of the largest, to see how exactly how homologous the ROI placement and size is from a morphological perspective. I think such tests/figures are particularly important since only one specimen represents the trabecular parameters for most of the study species.

In the results section, I am a bit confused by the first sentence (pg. 8): “The lifestyles represented by the mustelids we sampled indicate that the ancestral condition for the clade was ostensibly fossorial, and that five transitions – to the generalist, natatorial, and scansorial specializations, and one reversal to fossoriality – have occurred during their evolutionary history (Fig. 1).” Are

the phylogenetic relationships among mustelids not known from genetic data already? Mustelids are outside my research field, so I do not know, but the first sentence suggests that the phylogeny is being created using the locomotor specialisations first, and then mapping the trabecular parameters onto that phylogeny, rather than onto the “true” phylogeny. Is that correct? Or are the locomotor specialisations mapped onto the existing “true” phylogeny, which is what seems to be depicted in Figure 1 (but it’s not clear). If it’s the latter, has this not been done before? I’m just wondering if the “five transitions” found here is also novel information, or if previous research has determined the evolutionary locomotor history across mustelids and that therefore, this is information that can be presented in the Introduction. Also, the phylogenetic analysis is described at the very end of the Methods section, but presented at the beginning of the Results section. Since the focus on the paper is on the trabecular structure, it makes sense (to me anyway) to describe the trabecular structure first?

The Results section needs re-working. It is challenging to follow, the language at times is a bit colloquial/grammatically incorrect, and lacks necessary detail in some places. Eg. pg. 9, line 26 “... with differences that natators’ Tb.Th values are even higher than those of other types”. Is this a significant difference? Are natatorials significantly higher than ALL the other locomotor groups, or just some of them? Same at line 29: “BS values are only found as higher in fossors than in generalists”. Is this significant? And should be written as “BS values are only significantly higher in fossors compared with generalists.” I think p values could be included within the text throughout where significant differences are found.

It is unclear to me in the Results section what the body mass results actually are and how they relate to the trabecular parameters specifically. More information in the legend of Figure 5 is needed to tell the reader exactly how this PCA tells us about body mass effects. Related to this, the Discussion states “There was no effect of size on BV.TV for all but one ROIs we investigated (Table 2),...” (pg 12, line 38) but I do not see body mass data in Table 2.

The Discussion starts with a comparison to other clades (e.g. primates and zenarthrans) before actually discussing the results of mustelids. It would be better to present the results of this study first, and then move to the broader comparative context. Importantly, the Discussion needs to include some acknowledge of the fact that 22 of your 29 species are represented by n=1. This paper balances taxonomic breadth of the sample with small size, so it’s understandable, but one needs to clearly acknowledge the limitations of such a small sample.

Additional comments:

The introduction does not include any information about what “long bones” are being studied. This should be clarified in the Introduction, since there are many more long bones in the skeleton than just the humerus and femur.

Throughout the paper, since this is a three-dimensional analysis of trabecular bone, “region of interest” (ROI), would be more accurately called a “volume of interest” (VOI).

Throughout the paper the “locomotor groups” are referred to variously as “specialisations” or “types” or “locomotor specialisations” or “groupings”. Please pick one and use it consistently.

pg 3, ln 51: for this general, fundamental statement it would be good to cite some of the more classic literature/researchers on this topic, such papers/books by Currey, Rubin, Lanyon, Frost, etc. Currently only one experimental study by Barak et al. (2011) is cited.

pg. 4, ln 3: “Tb.Sp” stands for “trabecular separation”, not “trabecular spacing”. The standard abbreviation is “BV/TV”, not “BV.TV”, since it’s a ratio.

pg. 4, ln 31: “greater” than what? Perhaps change to simply “resist HIGH mechanical loads”

pg. 5, line 10: change “good enough” to “sufficient”

pg. 5, line 15: The statement “only right bones were used” is not true if you also used left bones and mirrored them. Please re-write, e.g. “Right bones were used when possible and, if not, the left bone was used and mirrored”.

pg. 5, line 17: Please clarify what is meant by “measurements”. What kind of measurements?

Reviewer: 2

Comments to the Author(s)

Dear Editor,

This study proposes to analyze the trabecular organization in the epiphyses of stylopod long bones in mustelids and notably focuses on the functional interpretations of the results. The study is carefully carried out and well presented. However, I have various points of concern.

I notably do not understand their body mass estimation. By making a regression between the mass taken from the literature and one parameter of their bone, they create a “rule” which suggests that the same relationships exists between this parameter and mass in all their taxa; the regression is phylogenetically informed, which can reduce this “overall rule”, correcting it by the phylogeny. However, to my knowledge none of the numerous body mass estimation formulae existing (from long bones [length, circumference.....], astragalus, teeth...) takes the phylogeny into account because proportions between a single skeletal element and mass have not so great chances to follow the phylogeny (although, for some of these formulae, some are restricted to some groups [e.g. artiodactyles, primates, carnivorans], which remains different than considering an impact of phylogeny). The approach sounds rather “heavy” in assumption for me. Moreover, if they have for one species a small sample, for another a giant one... it will add biases. I do not see why they do not rather estimate body mass using one of the numerous formulae; considering they have the stylopod long bones they can easily estimate it based on their two circumferences. In addition, here the parameter taken seems to be TV, which represents the total volume of the ROI, and which is thus subjective, not homologous between bones... BS would appear more easily “homologous”. I thus strongly recommend to estimate mass differently.

I also do not understand the way the authors deal with size for some parameters. If a correlation between size and the absolute thickness of the trabeculae naturally exists, I do not understand how they deal with this size effect. They do a linear regression and work on the residuals. This is often done for geometric morphometrics data when one variable does not necessarily correspond truly to one anatomical feature. However, here again, they thus consider a linear relationships between size and this parameter in all their sample. The “rule” is sample dependent of course. The correlation can vary between taxa. I do not understand why they cannot for example transform this parameter (in mm) influenced by size by making it a ratio; e.g. by dividing it by a size parameter (as homologous parameter as possible for size). This does not impose a similar relationship between size and this parameter (Tb.Th) but usually corrects the size effect, though some allometry can remain. If the authors prefer their method, could they justify why.

I also do not understand the inter-limb ratios. If the authors want to compare the structure of the humeral and femoral heads, dividing all (raw) parameters of one ROI by those of the other ROI appears very weird to me. Moreover, are the ROI of similar size?? The bones are not. This can have consequences. They could maybe do analyses of covariation like RV tests.

A more minor point, the authors say that collinear variables are excluded from the PCA; please justify this choice; and, especially, please say which ones (I do not recall covariation parameters in annex but maybe they are available). Also, for the PCA visualization, it would be great to add the variables on the graphs, and thus to see the contribution of each variable.

About the material... determining categories is awful, because of course there are continuums. But it is necessary. However, *Mustela lutreola*, though semi-aquatic, appears adapted to a much lesser extent to an aquatic locomotion than otters. I think it might bias results considering the aquatic group. Did you try without this taxon in this category; it is probably much more different from other categories. Because the anatomy of *M. lutreola* is poorly distinct from that of other mustelines.

In the discussion, it would be great to add details about the way you compare the ranges of variation between taxa in your study and data from other studies on other animals/bones. Moreover, the ROIs differ between all these analyses. Please be more specific in the comparisons. Also, please clarify between primates and anthropoids so that the reader really understands which type of primates you are talking about in each case.

I also added a few minor comments directly in the pdf. Please indicate the reference for the phylogeny on figure 1.

After consideration of these comments and a few changes, this study will be a nice contribution, adding data concerning mustelids to a for now still small data basis about the 3D trabecular organization in limb long bones of various amniotes.

I remain at your disposal if you have any comment or question.

Best wishes,

Alexandra Houssaye

Author's Response to Decision Letter for (RSOS-190938.R0)

See Appendix A.

RSOS-190938.R1 (Revision)

Review form: Reviewer 1

Is the manuscript scientifically sound in its present form?

Yes

Are the interpretations and conclusions justified by the results?

Yes

Is the language acceptable?

Yes

Do you have any ethical concerns with this paper?

No

Have you any concerns about statistical analyses in this paper?

No

Recommendation?

Accept with minor revision (please list in comments)

Comments to the Author(s)

Please see attached PDF (Appendix B).

Review form: Reviewer 3

Is the manuscript scientifically sound in its present form?

No

Are the interpretations and conclusions justified by the results?

No

Is the language acceptable?

Yes

Do you have any ethical concerns with this paper?

No

Have you any concerns about statistical analyses in this paper?

No

Recommendation?

Major revision is needed (please make suggestions in comments)

Comments to the Author(s)

Comments to the author are sent in the attach file below (Appendix C).

Decision letter (RSOS-190938.R1)

08-Sep-2019

Dear Dr Amson:

On behalf of the Editors, I am pleased to inform you that your Manuscript RSOS-190938.R1 entitled "Trabecular bone architecture in the stylopod epiphyses of mustelids (Mammalia, Carnivora)" has been accepted for publication in Royal Society Open Science subject to minor revision in accordance with the referee suggestions. Please find the referees' comments at the end of this email.

The reviewers and Subject Editor have recommended publication, but also suggest some minor

revisions to your manuscript. Therefore, I invite you to respond to the comments and revise your manuscript.

- Ethics statement

- Data accessibility

If you wish to submit your supporting data or code to Dryad (<http://datadryad.org/>), or modify your current submission to dryad, please use the following link:
<http://datadryad.org/submit?journalID=RSOS&manu=RSOS-190938.R1>

- Competing interests

- Authors' contributions

- Acknowledgements

- Funding statement

Please note that we cannot publish your manuscript without these end statements included. We

have included a screenshot example of the end statements for reference. If you feel that a given heading is not relevant to your paper, please nevertheless include the heading and explicitly state that it is not relevant to your work.

Because the schedule for publication is very tight, it is a condition of publication that you submit the revised version of your manuscript before 17-Sep-2019. Please note that the revision deadline will expire at 00.00am on this date. If you do not think you will be able to meet this date please let me know immediately.

Kind regards,
Andrew Dunn
Royal Society Open Science Editorial Office
Royal Society Open Science

on behalf of Dr Michael Doube (Associate Editor) and Kevin Padian (Subject Editor)
openscience@royalsociety.org

Associate Editor Comments to Author (Dr Michael Doube):

Associate Editor: 1

Comments to the Author:

Dear Dr Amson,

Thank you for revising your manuscript in response to the reviewers' comments.

Both reviewers have suggestions that can be handled by alterations to the text of your manuscript. In particular, please justify your choice of VOI location and its potential biasing effect on the measured trabecular parameters, due to sampling trabecular variation in a restricted region within the epiphyses. You may if you wish repeat your inter-observer error measurements according to the reviewer's recommendation. It would also be helpful to update Fig 2 as R2 suggests to more clearly indicate the location of the VOIs. I am less concerned about the body size estimation than previously, however, R2's continued concern should be addressed by a clear description and defence of the method in the methods section. Please refer to Fig 1. early in the Methods section where you refer to the timetree for the first time.

P4 L46 defers -> differs

I look forward to receiving your revised manuscript.

Reviewer comments to Author:

Reviewer: 1

Comments to the Author(s)

Please see attached PDF

Reviewer: 3

Comments to the Author(s)

Comments to the author are sent in the attach file below.

Author's Response to Decision Letter for (RSOS-190938.R1)

See Appendix D.

Decision letter (RSOS-190938.R2)

20-Sep-2019

Dear Dr Amson,

I am pleased to inform you that your manuscript entitled "Trabecular bone architecture in the stylopod epiphyses of mustelids (Mammalia, Carnivora)" is now accepted for publication in Royal Society Open Science.

on behalf of Dr Michael Doube (Associate Editor) and Kevin Padian (Subject Editor)
openscience@royalsociety.org

Follow Royal Society Publishing on Twitter: [@RSocPublishing](https://twitter.com/RSocPublishing)

Appendix A

MUSEUM FÜR NATURKUNDE BERLIN · INVALIDENSTRASSE 43 · 10115 BERLIN

Leibniz-Institut für Evolutions-
und Biodiversitätsforschung

Dr. Eli Amson
DFG postdoctoral researcher

Tel +49 30 889140 8339
Mobil +49 152 3477 5360
Mail Eli.Amson@mfn.berlin

www.museumfuernaturkunde.berlin

Response to Referrees

Berlin, 15 July 2019

Dear Editor,

We have carefully considered all recommendations and criticisms of the Associate Editor and reviewers. We trust that the revised manuscript (MS) is clearer and more rigorous. Furthermore, we added four additional Supplementary Online Material documents. Please find below a detailed response to all comments. Along with the clean main document, we provided a MS Word with tracked changes, in order to signal all the edits that were performed, as well as an annotated pdf file to answer comments made on such a file by one of the Reviewers.

Associate Editor's comments (Dr Michael Doube):

Dear Dr Amson,

Two reviewers have given detailed recommendations and made several well-founded criticisms of this manuscript. I agree with the concerns around estimating body mass from a bone parameter, when the relation between body mass and the bone parameter is likely to vary by phylogeny, and potentially with the lifestyle categories you have selected. It would be worthwhile identifying where your interpretations rely on assumptions or error-prone interpolation and move towards using direct measurements rather than derived terms for your analysis.

Please revise the manuscript, and in your response list a reply to all of the reviewers' comments.

Kind regards,
Michael Doube

We have carefully examined the recommendations of the two Reviewers, some of which are summarized by the Associate Editor. Please see below how we have addressed these recommendations in detail. The concern mentioned by the Associate Editor is related to the inclusion of body size in the analysis. Some authors favour the approach of using a whole-body metric (which is arguably particularly relevant biomechanically), such as body mass. However, collection specimens are rarely associated with body mass data, so authors usually resort in using species mean from databases (e.g., Fajardo et al. 2013). In order to have a measurement directly related to the studied specimen, some authors alternatively choose to directly use a specimen measurement that is expected to

be a good proxy for body size (e.g., Houssaye et al. 2016). A third option commonly used (often in analyses comprising extinct taxa) is to use a specimen measurement to estimate body size (e.g., (Saers et al. 2019)). We are not aware of any study systemically checking which of the three possible approaches is the most appropriate. For the present work we favour the latter, which can be argued to represent a whole-body metric specific to the studied specimens. This is explained further in the response below. As we argue below, phylogeny must be involved for this procedure. But we have demonstrated in this Response that the differences with an 'ordinary' analysis are negligible. Furthermore, as explained below, the specimen-specific measurement we used corresponds to the size of the Region of Interest, which is defined in such a way (see Methods and SOM 3) that it can be viewed as a direct measurement relating to the size of the studied articular structure.

Comments to Author:

Reviewers' Comments to Author:

Reviewer: 1

Comments to the Author(s)

This paper quantifies the trabecular structure within the proximal and distal epiphyses of the humerus and femur in mustelids. They sample 36 bones across a broad sample of 29 species, which they divide into four locomotor groups: natatorial, fossorial, scansorial and generalists. They look at the relationship between trabecular parameters and variation in body mass, and how trabecular parameters vary across the mustelid phylogeny. They find limited evidence for differences in trabeculae across the different locomotor groups, apart from higher BV/TV in natatorial species.

Overall, this is an interesting study and provide novel data on an interesting group mammals that vary substantially in the locomotor behaviours. The authors provide a good broader context on how mustelid trabecular morphology compares to other mammals. I like that the authors have considered variation in body mass and the phylogenetic relationships among the study taxa when analysing/interpreting their data. The writing within the Introduction especially, and generally the Discussion is clear and concise. I have, however, a few major concerns about hypotheses and methods and there were several points where it was unclear to me what was being done or why. I have outlined these, as well as more minor comments, below. Mustelids are not my field of expertise, which likely added some additional confusion that would not be expected from an expert. However, given the broad readership of this journal, it is worth defining terminology or explaining predictions (e.g. greater bone for buoyancy) that non-experts may not be familiar with. First and foremost, I would recommend defining "natatorial", "fossorial", and "scansorial".

Definitions added to the revised MS.

Also, there are a LOT of abbreviations in this manuscript. Removing some of them will make it much more reader-friendly.

We agree that abbreviations are sometimes not reader-friendly. But most abbreviations we used are standard ones defining trabecular parameters, so it would be difficult to avoid them. The other common abbreviation used in the MS is “ROI” (for “region of interest”), which we believe is very commonly used in many fields.

I find the study predictions (pg 4) difficult to follow. It is not clear to me why natatorial mustelids are predicted to have greater BV/TV, TbTh and Conn.D, and lower Tb.Sp when, as aquatic tetrapods, buoyancy is considered to be the most important functional constraint. I would think less bone, rather than more, might be better for buoyancy? (see comment above). The prediction about natatorial mustelids seems to be the same as that of fossorial taxa, which are also predicted to have greater BV/TV, TbTh, Conn.D and lower Tb.Sp (lines 19-20). How do you expect that the fossorial and natatorial taxa will compare/differ?

We understand this concern, as bone density and buoyancy are associated in two opposite adaptations: aquatic tetrapods swimming in shallow water need to counteract buoyancy by increasing their density (like the lead belt of scuba divers), hence our expectations for mustelids. On the contrary active swimmers diving at great depth (e.g., extant cetaceans) do show the opposite trend, with some bone compactness reduction. We edited the manuscript to make this clearer.

The initial predictions we’ve made about fossorial and natatorial mustelids were not the same, as in the former case we were referring to the forelimb ROIs, while in the latter case we referred to all studied ROIs (including those of the hindlimb). We made that clearer in the revised MS.

When one gets to the Methods, it is clear that bone surface area (BS) and mean direction of the trabeculae (MDT) are also quantified, but they are not included in the predictions. Why not? Are these variables adding any useful information/results that are not already quantified in the other parameters? Maybe so, but just worth considering...just because they are quantified in BoneJ doesn’t mean that they all need to be included. In particular, predictions about MDT could be made based on difference in joint angles or joint load perhaps? Currently, MDT is reported in the results as showing no clear differences across the locomotor groups, but without any context as to whether this is an expected or surprising results. Either link MDT to something specific about the locomotor groups, or leave it out.

We do believe that the MDT adds information not captured by the other parameters, as it is the only parameter that relates to the anatomical direction of the trabeculae. It is true that prediction regarding this parameter could be made using posture/joint angle data. But, to our knowledge, such data is not

systematically available yet. We hence prefer to keep the descriptive data about MDT, in the eventuality of future works acquiring such data. We hence moved the section and corresponding figure to the Supplementary Online Material.

We agree that it is not because a parameter is implemented in BoneJ that it should necessarily be investigated. Many other parameters implemented in BoneJ were not presented herein. While we don't have specific expectations for BS, we decided to investigate this parameter as it was found in previous analyses as discriminating functional groups (e.g., Ryan and Shaw 2015; Scherf et al. 2013; therein a ratio to BV and/or TV). This was added to the revised MS.

The Methods describe "This macro was used to extract a substack centred in each epiphysis that is as big as possible without including cortical bone. The epiphyses of some small-sized mustelids only contained a few trabeculae. We therefore chose this approach to acquire the 'bulk' of the epiphyseal trabeculae." I understand the constraints of size and morphological variation across different taxa and thus why the authors have chosen this method. However, I would like to see some intra/interspecific assessment of how well this method works and how well it avoids the over-sampling problem (see Fajardo and Muller, 2001 *Am J Phys Anthropology*). For example, were intra/inter-observer error tests conducted to see if the ROI could be placed in the same position over multiple trials or how much the trabecular parameters varied with slight differences in ROI position? Furthermore, the ROI in the proximal humerus and, less so, the femoral condyle, is very small compared to the size of the epiphysis/amount of trabeculae that is NOT quantified so I don't think this can be described as a "bulk" approach. ROI size is particularly important when one gets to the Discussion, with statements like "The ROIs were selected to comprise the bulk of the epiphysis' trabeculae, and are hence expected to be scale proportionally to the size of the epiphysis (see proximal interlimb ratio, SOM 1).", in which the authors are specifically referring to the humeral and femoral heads. The relative amount of trabeculae being quantified in each is very different, according to Figure 2.

We have followed this recommendation and added to the revised MS an assessment of the intra-observer error (inter-observer assessment was not performed because only one of the authors took part in the data acquisition, which should ideally be only performed by a single operator). We consider that the errors over 10 repeats were satisfactory, the coefficients of variation of the parameters ranging from 0.14 to 4.71% (mean cv = 2.19%).

Differences in the position of ROI were shown to influence trabecular parameters indeed, as described by Kivell et al. (2011). While we do not consider that re-evaluating this for our dataset is necessary, we made a mention of it in the revised MS.

The Reviewer correctly identified an imprecise formulation of ours: what we referred to as "size of the epiphysis" in the quoted sentence (and similar sentences throughout the MS) should more accurately be referred to as the size of the articular structure in question, e.g., humeral head. Accordingly, we now refrain from referring to the "bulk" of the trabeculae. We have edited the MS to correct

these imprecisions. We do not completely agree with the statement that the relative amount of trabeculae quantified in humeral and femoral heads is very different, because it should not be considered that all the trabeculae of the proximal epiphysis of the humerus (see Fig. 2A) pertain to the humeral head. In relation to this, we do not deem necessary to demonstrate once again the “over-sampling problem”, but we added a reference to it in the revised MS.

Figure 2 is very useful, but it would be helpful to see an example of the ROI placement in one of the smallest taxa and one of the largest, to see how exactly how homologous the ROI placement and size is from a morphological perspective. I think such tests/figures are particularly important since only one specimen represents the trabecular parameters for most of the study species.

The SOM 3 (new numbering) was added to show the ROI definition for the smallest and largest taxa of the dataset.

In the results section, I am a bit confused by the first sentence (pg. 8): “The lifestyles represented by the mustelids we sampled indicate that the ancestral condition for the clade was ostensibly fossorial, and that five transitions – to the generalist, natatorial, and scansorial specializations, and one reversal to fossoriality – have occurred during their evolutionary history (Fig. 1).” Are the phylogenetic relationships among mustelids not known from genetic data already? Mustelids are outside my research field, so I do not know, but the first sentence suggests that the phylogeny is being created using the locomotor specialisations first, and then mapping the trabecular parameters onto that phylogeny, rather than onto the “true” phylogeny. Is that correct? Or are the locomotor specialisations mapped onto the existing “true” phylogeny, which is what seems to be depicted in Figure 1 (but it’s not clear). If it’s the latter, has this not been done before? I’m just wondering if the “five transitions” found here is also novel information, or if previous research has determined the evolutionary locomotor history across mustelids and that therefore, this is in information that can be presented in the Introduction. Also, the phylogenetic analysis is described at the very end of the Methods section, but presented at the beginning of the Results section. Since the focus on the paper is on the trabecular structure, it makes sense (to me anyway) to describe the trabecular structure first?

The first sentence of the Results section was re-written to clarify the doubts raised here.

We have decided to describe the results of the phylogenetic reconstruction of the locomotor types transitions at the beginning of the Results section because this is required to assess whether or not the locomotor types are clustered on the phylogeny. This, in turn, is required to interpret the rest of the results, i.e, the phylogenetically informed AN(CO)VAs. We hence think it is more appropriate to maintain this organisation for the Results section.

The Results section needs re-working. It is challenging to follow, the language at times is a bit colloquial/grammatically incorrect, and lacks necessary detail in some places.

Eg. pg. 9, line 26 "... with differences that natators' Tb.Th values are even higher than those of other types". Is this a significant difference? Are natatorials significantly higher than ALL the other locomotor groups, or just some of them? Same at line 29: "BS values are only found as higher in fossors than in generalists". Is this significant? And should be written as "BS values are only significantly higher in fossors compared with generalists." I think p values could be included within the text throughout where significant differences are found.

The Results section was revised in order to be less colloquial and more rigorous grammatically. P-values were included within the text of the revised MS as recommended.

It is unclear to me in the Results section what the body mass results actually are and how they relate to the trabecular parameters specifically. More information in the legend of Figure 5 is needed to tell the reader exactly how this PCA tells us about body mass effects. Related to this, the Discussion states "There was no effect of size on BV.TV for all but one ROIs we investigated (Table 2),..." (pg 12, line 38) but I do not see body mass data in Table 2.

We are not sure what the Reviewer means by "the body mass results". The correlation of the body mass with each parameter (and for each ROI) is given in the Table 3 and SOM 9 (SOM 6 in the initial submission). The PCA does not strictly assess the body mass effects, one can only assess the loading of the body mass when it is included in the PCA as a variable. As suggested by the Reviewer, we edited the legend of Figure 5 to clarify the matter. The Reviewer is right in identifying that, in the quoted sentence, the wrong Table was referred to. This is corrected in the revised MS.

The Discussion starts with a comparison to other clades (e.g. primates and zenarthrans) before actually discussing the results of mustelids. It would be better to present the results of this study first, and then move to the broader comparative context. Importantly, the Discussion needs to include some acknowledge of the fact that 22 of your 29 species are represented by n=1. This paper balances taxonomic breadth of the sample with small size, so it's understandable, but one needs to clearly acknowledge the limitations of such a small sample.

The Discussion was edited to start with the results about mustelids. The revised MS also acknowledges the limitation of a low sample size on the estimation of interspecific variation.

Additional comments:

The introduction does not include any information about what "long bones" are being studied. This should be clarified in the Introduction, since there are many more long bones in the skeleton than just the humerus and femur.

Added to the revised MS.

Throughout the paper, since this is a three-dimensional analysis of trabecular bone, “region of interest” (ROI), would be more accurately called a “volume of interest” (VOI).

Replaced throughout the revised MS.

Throughout the paper the “locomotor groups” are referred to variously as “specialisations” or “types” or “locomotor specialisations” or “groupings”. Please pick one and use it consistently.

To comply with this recommendation, the whole MS was edited to strictly refer to the four “locomotor types”, unless it was one of the three locomotor specializations (natatorial, fossorial, scansorial) that was the intended meaning. We used “groupings” only once, to define the concept of the analysis (comparing group differences). We prefer to keep this formulation as is for clarity.

pg 3, ln 51: for this general, fundamental statement it would be good to cite some of the more classic literature/researchers on this topic, such papers/books by Currey, Rubin, Lanyon, Frost, etc. Currently only one experimental study by Barak et al. (2011) is cited.

References added to the revised MS.

pg. 4, ln 3: “Tb.Sp” stands for “trabecular separation”, not “trabecular spacing”. The standard abbreviation is “BV/TV”, not “BV.TV”, since it’s a ratio.

Corrected in the revised MS.

pg. 4, ln 31: “greater” than what? Perhaps change to simply “resist HIGH mechanical loads”

Changed in the revised MS.

pg. 5, line 10: change “good enough” to “sufficient”

Changed in the revised MS.

pg. 5, line 15: The statement “only right bones were used” is not true if you also used left bones and mirrored them. Please re-write, e.g. “Right bones were used when possible and, if not, the left bone was used and mirrored”.

Corrected in the revised MS.

pg. 5, line 17: Please clarify what is meant by “measurements”. What kind of measurements?

Clarified in the revised MS.

Reviewer: 2

Comments to the Author(s)

Dear Editor,

This study proposes to analyze the trabecular organization in the epiphyses of stylopod

long bones in mustelids and notably focuses on the functional interpretations of the results. The study is carefully carried out and well presented. However, I have various points of concern.

I notably do not understand their body mass estimation. By making a regression between the mass taken from the literature and one parameter of their bone, they create a “rule” which suggests that the same relationships exists between this parameter and mass in all their taxa; the regression is phylogenetically informed, which can reduce this “overall rule”, correcting it by the phylogeny. However, to my knowledge none of the numerous body mass estimation formulae existing (from long bones [length, circumference.....], astragalus, teeth...) takes the phylogeny into account because proportions between a single skeletal element and mass have not so great chances to follow the phylogeny (although, for some of these formulae, some are restricted to some groups [e.g. artiodactyles, primates, carnivorans], which remains different than considering an impact of phylogeny). The approach sounds rather “heavy” in assumption for me. Moreover, if they have for one species a small sample, for another a giant one... it will add biases. I do not see why they do not rather estimate body mass using one of the numerous formulae; considering they have the stylopod long bones they can easily estimate it based on their two circumferences. In addition, here the parameter taken seems to be TV, which represents the total volume of the ROI, and which is thus subjective, not homologous between bones... BS would appear more easily “homologous”. I thus strongly recommend to estimate mass differently.

We strongly disagree with the Reviewer opinion that the regression used to estimate body mass should never be phylogenetically informed. It’s not really clear what is meant by “can reduce this “overall rule””. As well explained in Garland, Jr. and Ives (2000), and contrary to what the Reviewer argues, predictions can be more accurate when the analysis is phylogenetically informed (see also recent example specifically estimating body mass with phylogenetic methods in De Esteban-Trivigno and Köhler 2011). In any case we use an approach that assesses the phylogenetic signal of the relationship in question (residuals of mass~TV). So, if as the Reviewer predicts, “proportions between a single skeletal element and mass have not so great chances to follow the phylogeny,” the lambda used to include the phylogeny of the regressions will be close to 0, and the regression will yield the same results as an ordinary regression. But, as a matter of fact, the recovered lambda values for the regressions mass~TV range from 0.37 to 0.96, (data not provided in the initial submission). This confirms that the correlation is not independent from the phylogeny. That being said, and even though we consider more rigorous to include phylogeny in this part of the analysis, we have compared the outcome of a phylogenetically informed mass estimation (following our methodology) to an ordinary estimation (where the phylogeny is disregarded). Below is a plot showing that the mass estimations are extremely similar (values are actually different):

Similarly, the outcome of the analysis using the phylogenetically non-informed mass estimation is almost identical (compare to Table 3):

ROI	Sc?	lambda	P	F
DA_HpBMspNo	FALSE	<0	0.529	0.74
Conn.D_HpBMspNo	FALSE	>1	0.002	9.33
BV.TV_HpBMspNo	FALSE	0.99	0.039	3.4
Tb.Th.Mean_HpBMspNo	FALSE	>1	0.002	7.68
Tb.Sp.Mean_HpBMspNo	TRUE	0.81	0.027	2.57
BS_HpBMspNo	TRUE	>1	0.001	0.64

By the statement “if they have for one species a small sample, for another a giant one... it will add biases”, we understand that Reviewer raises the potential bias involved by sampling only a few (or even only one) specimen per species. For our study we trust that our sampling is not unreasonable or biased, as the groupings in the figures do not appear to be plagued by consistent and notable outliers. Also, in a previous study that mostly used the same specimens (Kilbourne and Hutchinson 2019), clear functional groupings were obtained. However, the revised MS acknowledges the limitation of a low sample size (see also comment of first Reviewer above).

We have decided to use TV as the parameter measured on each specimen, because it directly relates to the size of the articular structure, as being defined as the largest cube centred in this articular structure. It is therefore as “subjective” as using the circumference of long bones at midshaft (which requires to define midshaft and the exact plane of the cross-section). Given the employed procedure (see SOM 3), the defined ROIs are structurally analogous (note that a homology criterion for a circumference of long bones at midshaft is neither straightforward nor warranted when interested in its biomechanical properties).

Finally, it does not make sense to us to use BS, which depends on the number and shape of trabeculae included in the ROI (which can be largely different for two species of similar sizes).

I also do not understand the way the authors deal with size for some parameters. If a correlation between size and the absolute thickness of the trabeculae naturally exists, I do not understand how they deal with this size effect. They do a linear regression and work on the residuals. This is often done for geometric morphometrics data when one variable does not necessarily correspond truly to one anatomical feature. However, here again, they thus consider a linear relationships between size and this parameter in all their sample. The “rule” is sample dependent of course. The correlation can vary between taxa. I do not understand why they cannot for example transform this parameter (in mm) influenced by size by making it a ratio; e.g. by dividing it by a size parameter (as homologous parameter as possible for size). This does not impose a similar relationship between size and this parameter (Tb.Th) but usually corrects the size effect, though some allometry can remain. If the authors prefer their method, could they justify why.

As clearly stated in the initial MS, and contrary to what the Reviewer states, we did not use the residuals (of a regression of the parameter ~ body mass) for the analyses. Again, as clearly stated in the initial MS, these residuals are only provided for a visual assessment (e.g., Fig 3). But we edited the MS to make that even clearer. The analyses take into account the effect of size (if present) by including body mass as a covariate (in the ANCOVAs), a widely used method.

I also do not understand the inter-limb ratios. If the authors want to compare the structure of the humeral and femoral heads, dividing all (raw) parameters of one ROI by those of the other ROI appears very weird to me. Moreover, are the ROI of similar size?? The bones are not. This can have consequences. They could maybe do analyses of covariation like RV tests.

Without letting us know why the approach “appears very weird”, it is difficult to address this comment. The ROIs are not of similar size. Again, we are curious of the “consequences” mentioned by the Reviewer, but cannot really grasp what they are. So we can only give an published example of a similar approach: Ryan & Walker (2010). This approach was mainly used to compare DA and BV/TV from the humeral and femoral heads...using a ratio simply tells if one of the heads is more anisotropic / dense than the other.

A more minor point, the authors say that collinear variables are excluded from the PCA; please justify this choice; and, especially, please say which ones (I do not recall covariation parameters in annex but maybe they are available). Also, for the PCA visualization, it would be great to add the variables on the graphs, and thus to see the contribution of each variable.

We have reconsidered this choice and now include all variables in the PCA (this implies that we have more variables than observations, but this should

not be a problem, as we only study the first three PCs). Because many variables are included, the PCA visualization suggested by the Reviewer is somewhat crowded. We have hence included such a Figure in the additional SOM 11.

About the material... determining categories is awful, because of course there are continuums. But it is necessary. However, *Mustela lutreola*, though semi-aquatic, appears adapted to a much lesser extent to an aquatic locomotion than otters. I think it might bias results considering the aquatic group. Did you try without this taxon in this category; it is probably much more different from other categories. Because the anatomy of *M. lutreola* is poorly distinct from that of other mustelines.

We agree that the locomotor categories defined for some mustelids are approximations, and that some taxa should probably have their own type, or belong to several types. We did not try to change the *a priori* assignments of taxa, or to re-run the analysis excluding some taxa. But we added to the Discussion of the revised MS a paragraph to tackle this issue.

In the discussion, it would be great to add details about the way you compare the ranges of variation between taxa in your study and data from other studies on other animals/bones. Moreover, the ROIs differ between all these analyses. Please be more specific in the comparisons. Also, please clarify between primates and anthropoids so that the reader really understands which type of primates you are talking about in each case.

The relevant section of the Discussion was edited to follow these recommendations in the revised MS. Because these are very rough comparisons (the Reviewer is right in stating that ROIs differ between some analyses), we prefer not to differentiate between non-anthropoid primates and anthropoids.

I also added a few minor comments directly in the pdf. Please indicate the reference for the phylogeny on figure 1.

The comments annotated on the pdf were included in the revised MS, or, if that was not the case, a response was added to each individual comment (see accompanying pdf). The reference of the phylogeny we used was already indicated in the initial submission (Material and Methods section). But maybe the Reviewer meant that we should indicated it in the Figure legend as well? For this eventuality we have added to the revised MS.

Sincerely yours, and on behalf of both co-authors,
Eli Amson

Cited references

De Esteban-Trivigno, S., & Köhler, M. (2011). New equations for body mass

- estimation in bovids: Testing some procedures when constructing regression functions. *Mammalian Biology*, 76(6), 755–761.
- Fajardo, R. J., Desilva, J. M., Manoharan, R. K., Schmitz, J. E., Maclatchy, L. M., & Boussein, M. L. (2013). Lumbar Vertebral Body Bone Microstructural Scaling in Small to Medium-Sized Strepsirhines. *Anatomical Record*, 296(2), 210–226.
- Garland, Jr., T., & Ives, A. R. (2000). Using the Past to Predict the Present: Confidence Intervals for Regression Equations in Phylogenetic Comparative Methods. *The American Naturalist*, 155(3), 346–364.
- Houssaye, A., Fernandez, V., & Billet, G. (2016). Hyperspecialization in Some South American Endemic Ungulates Revealed by Long Bone Microstructure. *Journal of Mammalian Evolution*, 23(3), 221–235.
- Kilbourne, B. M., & Hutchinson, J. R. (2019). Morphological diversification of biomechanical traits: Mustelid locomotor specializations and the macroevolution of long bone cross-sectional morphology. *BMC Evolutionary Biology*, 19(1), 1–16.
- Kivell, T. L., Skinner, M. M., Lazenby, R., & Hublin, J.-J. (2011). Methodological considerations for analyzing trabecular architecture: an example from the primate hand. *Journal of anatomy*, 218(2), 209–25.
- Ryan, T. M., & Shaw, C. N. (2015). Gracility of the modern *Homo sapiens* skeleton is the result of decreased biomechanical loading. *Proceedings of the National Academy of Sciences*, 112(2), 372–377.
- Ryan, T. M., & Walker, A. (2010). Trabecular bone structure in the humeral and femoral heads of anthropoid primates. *Anatomical record*, 293(4), 719–29.
- Saers, J. P. P., Ryan, T. M., & Stock, J. T. (2019). Trabecular bone functional adaptation and sexual dimorphism in the human foot. *American Journal of Physical Anthropology*, 168(1), 154–169.
- Scherf, H., Harvati, K., & Hublin, J. (2013). A comparison of proximal humeral cancellous bone of great apes and humans. *Journal of Human Evolution*, 65(1), 29–38.

Appendix B

The authors have made a concerted effort to address and/or incorporate all of the comments from both reviewers and the associate editor. In particular, aspects of the hypotheses and methods that I found initially confusing are now much more clear. The results sections is also easier to follow and I appreciate the efforts that went into re-writing this section. Overall, I am generally happy with the changes made or the response provided by the author's when a suggestion was not incorporated.

However, I have a few more substantial and some minor comments below that I think still need to be addressed, in which I refer below to the page/line numbers of the "tracked changes" version of the revised manuscript.

Regarding my previous comment about the difference in the relative amount of trabecular bone quantified within the humeral vs femoral head, the author's response was: "*We do not completely agree with the statement that the relative amount of trabeculae quantified in humeral and femoral heads is very different, because it should not be considered that all the trabeculae of the proximal epiphysis of the humerus (see Fig. 2A) pertain to the humeral head. In relation to this, we do not deem necessary to demonstrate once again the "oversampling problem", but we added a reference to it in the revised MS.*" I think what the authors are aiming to articulate here and in the change from "epiphysis" to "articular structure" is the "articular surface". I agree that all trabeculae within the humeral head/epiphyses are not necessarily (potentially) reflecting load from the articular surface with the glenoid cavity. However, part of my original critique was that – based on the images in Figure 2 (see below) and in the SOM3 – the VOI in the femoral head and, less so the lateral condyle, is positioned within the centre, further away from the articular surface and quantifying a substantial proportion of the trabeculae within this epiphysis. In contrast, the VOI in the humeral head is positioned directly under the articular surface and there is comparatively much more of the trabecular structure, not just within the head/shaft of the humerus, but also under the articular surface that is NOT quantified by the VOI.

Related to this, the placement of the humeral head VOI is described as:

"The humeral head VOI was bounded anterolaterally by the maximum concavity of the lateral side of head, medially by medial-most point at the level of the anterolateral corner (just defined), and posteriorly by the posterior-most level of head (note that the VOI does not appear as centred in Fig. 2A because the anteroposterior depth of the articular surface at mid-length does not extend as far anteriorly as the anteriormost edge of the surface visible of the figure)." I assume (although I am not familiar with mustelid humeral morphology) from this description the boundaries are defined not on the articular surface itself, but on non-articular areas of the humeral head. Of course, it is important that these boundaries/landmarks are comparable, identifiable, and homologous across the different morphologies of all of the sample taxa. However, as mentioned above, the humeral head and humeral trochlea appear to sample trabecular directly under the articular surface while the VOI in the femoral head, and less so the lateral condyle, are sampling more internal/central trabecular. Thus, one may be picking up slightly different biomechanical signals across these elements. SOM3 is certainly helpful for visualising the VOI size/placement/relative quantification of trabecular between the smallest and largest specimens – I appreciate this addition and think it has helped to improve the clarity of the methods. However, I think a potential bias/limitation of the VOI placement and the potential for slightly different biomechanical signals that the VOIs

are picking up between the humerus and femur should be acknowledged. The VOI placement may be fully justified by the variation in the size/morphology across the sample and ensuring the best possible homology and repeatability, but this limitation can still be acknowledged.

It is also not explained in the manuscript WHY these four VOIs were chosen. E.g. Why just the lateral femoral condyle and not the medial? The biomechanical and/or morphological reason for these VOIs should be provided in the Methods.

pg 7: I appreciate that the authors have added a test of intra-observer error. However, this test is on one specimen, one element, and repeated 10 times (and 10 times in a row? Or on different days? This information is not given). The point I made in my previous review is that VOI placement and homology is particularly important when there is an $n=1$ for most of the sample taxa (22 of 29 species). Thus a more robust test would be repeating the VOI placement multiple times (e.g. 3 times) across 5 or 10 different specimens/elements of varying size/morphology, to evaluate how repeatable the data are across different morphologies and how representative they are for taxa with low sample sizes.

pg 3, line 60: "Volume Of Interest" should not be capitalised (also in figure legends)

pg 4, line 40: change to "However, recent studies OF TETRAPODS SWIMMING IN SHALLOW WATER HAVE SHOWN THE NEED TO counteract..."

Appendix C

Review “Trabecular bone architecture in the stylopod epiphyses of mustelids (Mammalia, Carnivora)”

The paper “Trabecular bone architecture in the stylopod epiphyses of mustelids (Mammalia, Carnivora)” quantified different parameters of the trabecular architecture of epiphyses of humeri and femora of various species in mustelids, in order to understand the biomechanical effects of their locomotion specialization. I found the paper interesting, even if I believe the study will be stronger by adding a much larger number of samples to their analysis. The general writing of the paper is clear and concise. The objectives are set and well defined.

I have nevertheless some problems with two major methodology used in this research: body mass estimation and inter-limb ratios, which I explain in the general comments below.

General comments:

- Concerning the choice of the VOI in the different articular surface (p6 line 19 to 50). Even if the authors explain clearly, in the text, the space limitations to define the VOI according to the geometry of the different articular surface, I will suggest them to add: different views (lateral, anterior...) in 3D, without virtually cutting the bones (as they did in Figure 2), of two or more humeral and femoral surfaces in supplementary infos. It will help better to understand how the limit of the VOI were imposed.

- **Inter-limb ratios.** The authors quantified the inter-limb ratios, by dividing the values of the proximal humerus with proximal femur. I do not think this quantification make sense, or at least without the little explanations give by the authors. 1) Dividing raw parameters obtained in VOI of different size will give random results, I guess, and cannot be compared between different samples or even locomotion types groups; VOI differing between humeral and femoral in a same specimen but also between all the samples. Maybe to compare the inter-limb, it is better to make scatterplots comparing paired proximal humeral and proximal femoral (for example) trabecular bone structural values to see if there is a correlation between the humerus and femur in any trabecular bone variable. 2) Moreover both epiphyses of these stylopods will have different biomechanical constraint, according to the locomotion types: an humeral head and femoral head from one animal we have not the same function.

I do not clearly understand these inter-limbs ratio, so it should be more clearly explained in the paragraph 2 page 7, why they are looking at these inter-limb ratios. Maybe the authors could put some references if it was already used in some previous papers.

- **Body mass estimation:** I have a problem with the body mass estimation that the authors used in their paper. They stated “This estimation was obtained, for each VOI, using a regression of the species mean body mass (the latter is taken from the global database of late Quaternary mammal, MOM v4.1; [44]; unit: g) against a measure taken directly on each specimen, TV (see above).”

I do believe that the TV is not homologous between the different bones and samples and depends mostly of the size of the VOI selected. It is difficult to say if it is really representative of the size (and then the body mass) of the animal. At least it should be tested for one or two specimen with

a known body mass. I found to use this value will include a bias in their analysis; compared to have taken the bone length, bone circumference or even femoral head diameter that are usually taken for body mass estimation.

Additional comments:

-p3 line 34 “Specifically among mustelids, three main locomotor specializations can be [...] being skilful swimmers”. The sentence is formulated in a weird way. I will maybe write like :”three main locomotor specialization: fossorial like badgers...”

-p3 line 59 “Bone fraction (BV/TV) – bone volume relative to total volume – reflects the density of the studied Volume Of Interest (VOI).” I will avoid to use the word density here. BV/TV is defined as the volume of mineralized bone per unit volume of the sample. So it is not really the same as the density of the studied VOI. Density is more a concept of mass unit than volume unit.

-p4 line 6. What about BS (bone surface) that the authors used and talked after in the paper? Should it not be defined as well in this paragraph?

-p5 line 10. The authors claim they have analyzed 36 specimens. But only 33 specimens are present in the SOM1 table. In the SOM1, there is a spelling mistake to Ambonyx who should be Amblonyx and raw parameters for one specimen of this species are missing. Raw parameters for specimen Martes zibellina is missing and for the specimen of Lontra longicaudis.

-p6 line 53. The authors should put which specimen they used for the intra-observer calculation instead of the collection number.

-p9 line 22: For the abbreviations explanation in the SOM6 labels, I found a bit confusing that for example Hd which is I guess Humerus distal, d standing for distal is equal to the humeral trochlea. Maybe the authors should add there that it does stand for distal and correspond to the trochlea. The same explanation should be add for the other articular surfaces. I will suggest maybe the authors to make a list of abbreviations in the text or as supplementary information.

-p14 line 36 A reference of the peak forces measured in the ferret is needed here.

-p17 line 12: The authors talk about the Tb.Th values with the reduction of the forelimb. What about the trabecular spacing and density as these parameters are sometimes correlated? Did the authors find any signals in these last parameters with the forelimb reduction?

Appendix D

MUSEUM FÜR NATURKUNDE BERLIN · INVALIDENSTRASSE 43 · 10115 BERLIN

Leibniz-Institut für Evolutions-
und Biodiversitätsforschung

Dr. Eli Amson
DFG postdoctoral researcher

Tel +49 30 889140 8339
Mobil +49 152 3477 5360
Mail Eli.Amson@mfn.berlin

www.museumfuernaturkunde.berlin

Response Letter

Berlin, 16 September 2019

Dear Editor,

We have carefully considered once again all recommendations and criticisms of the Associate Editor and reviewers. Please find below a detailed response to all comments. We provide the manuscript revised for a second time (**RM**) as a clean main document and as a MS Word document with tracked changes.

Associate Editor's comments (Dr Michael Doube):

Dear Dr Amson,

Thank you for revising your manuscript in response to the reviewers' comments.

Both reviewers have suggestions that can be handled by alterations to the text of your manuscript. In particular, please justify your choice of VOI location and its potential biasing effect on the measured trabecular parameters, due to sampling trabecular variation in a restricted region within the epiphyses. You may if you wish repeat your inter-observer error measurements according to the reviewer's recommendation. It would also be helpful to update Fig 2 as R2 suggests to more clearly indicate the location of the VOIs. I am less concerned about the body size estimation than previously, however, R2's continued concern should be addressed by a clear description and defence of the method in the methods section. Please refer to Fig 1. early in the Methods section where you refer to the timetree for the first time.

We have addressed all Reviewer's suggestions, please see below.

P4 L46 defers -> differs

Corrected in the RM.

Reviewer: 1

The authors have made a concerted effort to address and/or incorporate all of the comments from both reviewers and the associate editor. In particular, aspects of the hypotheses and methods that I found initially confusing are now much more clear. The results sections is also easier to follow and I appreciate the efforts that went into re-writing this section. Overall, I am generally happy with the changes made or the response provided by the author's when a suggestion was not incorporated.

However, I have a few more substantial and some minor comments below that I think still need to be addressed, in which I refer below to the page/line numbers of the "tracked changes" version of the revised manuscript.

Regarding my previous comment about the difference in the relative amount of trabecular bone quantified within the humeral vs femoral head, the author's response was: *"We do not completely agree with the statement that the relative amount of trabeculae quantified in humeral and femoral heads is very different, because it should not be considered that all the trabeculae of the proximal epiphysis of the humerus (see Fig. 2A) pertain to the humeral head. In relation to this, we do not deem necessary to demonstrate once again the "oversampling problem", but we added a reference to it in the revised MS."* I think what the authors are aiming to articulate here and in the change from "epiphysis" to "articular structure" is the "articular surface". I agree that all trabeculae within the humeral head/epiphyses are not necessarily (potentially) reflecting load from the articular surface with the glenoid cavity. However, part of my original critique was that – based on the images in Figure 2 (see below) and in the SOM3 - the VOI in the femoral head and, less so the lateral condyle, is positioned within the centre, further away from the articular surface and quantifying a substantial proportion of the trabeculae within this epiphysis. In contrast, the VOI in the humeral head is positioned directly under the articular surface and there is comparatively much more of the trabecular structure, not just within the head/shaft of the humerus, but also under the articular surface that is NOT quantified by the VOI.

We do not consider that the femoral head VOI is much more internal/central than the other VOIs, as all VOIs are located directly under the articular surface along one of its sides (as a direct

consequence of the method of VOI placement). However, following this recommendation and that of the other Reviewer (see below) and Associate Editor, we have acknowledged this issue in the Material and Methods section.

Related to this, the placement of the humeral head VOI is described as:

The humeral head VOI was bounded anterolaterally by the maximum concavity of the lateral side of head, medially by medial-most point at the level of the anterolateral corner (just defined), and posteriorly by the posterior-most level of head (note that the VOI does not appear as centred in Fig. 2A because the anteroposterior depth of the articular surface at mid-length does not extend as far anteriorly as the anteriormost edge of the surface visible of the figure)." I assume (although I am not familiar with mustelid humeral morphology) from this description the boundaries are defined not on the articular surface itself, but on non-articular areas of the humeral head. Of course, it is important that these boundaries/landmarks are comparable, identifiable, and homologous across the different morphologies of all of the sample taxa. However, as mentioned above, the humeral head and humeral trochlea appear to sample trabecular directly under the articular surface while the VOI in the femoral head, and less so the lateral condyle, are sampling more internal/central trabecular. Thus, one may be picking up slightly different biomechanical signals across these elements. SOM3 is certainly helpful for visualising the VOI size/placement/relative quantification of trabecular between the smallest and largest specimens – I appreciate this addition and think it has helped to improve the clarity of the methods. However, I think a potential bias/limitation of the VOI placement and the potential for slightly different biomechanical signals that the VOIs are picking up between the humerus and femur should be acknowledged. The VOI placement may be fully justified by the variation in the size/morphology across the sample and ensuring the best possible homology and repeatability, but this limitation can still be acknowledged.

The boundaries are defined on the articular surface itself for all VOIs, including the humeral head. But we agree that the potential bias/limitation of the VOI placement should be acknowledged. Acknowledgement of the relative proportion of trabeculae included in the ROI, as well as the distance of the sampled trabeculae from the articular surface, are added to the RM.

It is also not explained in the manuscript WHY these four VOIs were chosen. E.g. Why just the lateral femoral condyle and not the medial? The biomechanical and/or morphological reason for these VOIs should be provided in the Methods. pg 7: I appreciate that the authors have added a test of intra-observer error. However, this test is on one specimen, one element, and repeated 10 times (and 10 times in a row? Or on different days? This information is not given). The point I made in my previous review is that VOI placement and homology is particularly important when there is an n=1 for most of the sample taxa (22 of 29 species). Thus a more robust test would be repeating the VOI placement multiple times (e.g. 3 times) across 5 or 10 different specimens/elements of varying size/morphology, to evaluate how repeatable the data are across different morphologies and how representative they are for taxa with low sample sizes.

As requested, we have added justification to our choice of VOIs. We agree that a more robust assessment of VOI placement error can be achieved by multiplying the number of specimens/observers, and by lengthening the time between the repeats. However, this is not a common practice in the field, and we deem it somewhat excessive. We have nevertheless added the requested precision about intra-observer error to the RM.

pg 3, line 60: "Volume Of Interest" should not be capitalised (also in figure legends)

Edited in the RM.

pg 4, line 40: change to "However, recent studies OF TETRAPODS SWIMMING IN SHALLOW WATER HAVE SHOWN THE NEED TO counteract...."

The sentence was modified in the RM.

Reviewer: 3

The paper "Trabecular bone architecture in the stylopod epiphyses of mustelids (Mammalia, Carnivora)" quantified different parameters of the trabecular architecture of epiphyses of humeri and femora of various species in mustelids,

in order to understand the biomechanical effects of their locomotion specialization. I found the paper interesting, even if I believe the study will be stronger by adding a much larger number of samples to their analysis. The general writing of the paper is clear and concise. The objectives are set and well defined. I have nevertheless some problems with two major methodology used in this research: body mass estimation and inter-limb ratios, which I explain in the general comments below.

General comments:

- Concerning the choice of the VOI in the different articular surface (p6 line 19 to 50). Even if the authors explain clearly, in the text, the space limitations to define the VOI according to the geometry of the different articular surface, I will suggest them to add: different views (lateral, anterior...) in 3D, without virtually cutting the bones (as they did in Figure 2), of two or more humeral and femoral surfaces in supplementary infos. It will help better to understand how the limit of the VOI were imposed.

Fig. 2 was completely remade to address this comment (see also above).

-Inter-limb ratios. The authors quantified the inter-limb ratios, by dividing the values of the proximal humerus with proximal femur. I do not think this quantification make sense, or at least without the little explanations give by the authors. 1) Dividing raw parameters obtained in VOI of different size will give random results, I guess, and cannot be compared between different samples or even locomotion types groups; VOI differing between humeral and femoral in a same specimen but also between all the samples. Maybe to compare the inter-limb, it is better to make scatterplots comparing paired proximal humeral and proximal femoral (for example) trabecular bone structural values to see if there is a correlation between the humerus and femur in any trabecular bone variable. 2) Moreover both epiphyses of these stylopods will have different biomechanical constraint, according to the locomotion types: an humeral head and femoral head from one animal we have not the same function. I do not clearly understand these inter-limbs ratio, so it should be more clearly explained in the paragraph 2 page 7, why they are looking at these inter-limb ratios. Maybe the authors could put some references if it was already used in some previous papers.

We strongly disagree with the assumption that “Dividing raw parameters obtained in VOI of different size will give random results.” It is not because two VOIs differ in size that for instance the mean thickness of the trabeculae they contain should necessarily vary, or for the trabecular architecture to be more or less anisotropic. Given the standard methodology we used to define the VOIs across the whole dataset, we deem our approach appropriate. Of course we agree that epiphyses have different function in an individual/species, and that’s precisely why

examining inter-limb ratio can be informative: they might reveal differential functions of the limbs in some of the studied taxa that can be overlooked by some systemic differences (Tsegai et al. 2018). We acknowledge the suggestion of the Reviewer to plot scatterplots of paired values. But this seems to eventually yield the same message. Ryan & Walker (2010) used a similar approach. This reference, along with other relevant ones, were already cited in the discussion of the inter-limbs ratios.

-Body mass estimation: I have a problem with the body mass estimation that the authors used in their paper. They stated "This estimation was obtained, for each VOI, using a regression of the species mean body mass (the latter is taken from the global database of late Quaternary mammal, MOM v4.1; [44]; unit: g) against a measure taken directly on each specimen, TV (see above)." I do believe that the TV is not homologous between the different bones and samples and depends mostly of the size of the VOI selected. It is difficult to say if it is really representative of the size (and then the body mass) of the animal. At least it should be tested for one or two specimen with a known body mass. I found to use this value will include a bias in their analysis; compared to have taken the bone length, bone circumference or even femoral head diameter that are usually taken for body mass estimation.

We have addressed this concern with clearer description and justification of the method in the Methods section.

Additional comments:

-p3 line 34 "Specifically among mustelids, three main locomotor specializations can be [...] being skilful swimmers". The sentence is formulated in a weird way. I will maybe write like: "three main locomotor specialization: fossorial like badgers..."

We disagree. The sentence was reviewed by a native English speaker.

-p3 line 59 "Bone fraction (BV/TV) – bone volume relative to total volume – reflects the density of the studied Volume Of Interest (VOI)." I will avoid to use the word density here. BV/TV is defined as the volume of mineralized bone per unit volume of the sample. So it is not really the same as the density of the studied VOI. Density is more a concept of mass unit than volume unit.

Corrected in the RM. We now refer to "bone fraction".

-p4 line 6. What about BS (bone surface) that the authors used and talked after in the paper? Should it not be defined as well in this paragraph?

Added to the RM.

-p5 line 10. The authors claim they have analyzed 36 specimens. But only 33 specimens are present in the SOM1 table. In the SOM1, there is a spelling

mistake to Ambonyx who should be Amblonyx and raw parameters for one specimen of this species are missing. Raw parameters for specimen *Martes zibellina* is missing and for the specimen of *Lontra longicaudis*.

Number of sampled specimens corrected in the RM and SOM. We did sample 35 specimens, but for some either the femur or the humerus was available, which explains the difference between the number of specimens and species sampled. The typo is corrected. It is unclear to us why the Reviewer did not find raw that for *Martes zibellina* and *Lontra longicaudis*. The former is represented in the SOM by two specimens, and the latter by one specimen, which missed the femur.

-p6 line 53. The authors should put which specimen they used for the intra-observer calculation instead of the collection number.

Added to the RM.

-p9 line 22: For the abbreviations explanation in the SOM6 labels, I found a bit confusing that for example Hd which is I guess Humerus distal, d standing for distal is equal to the humeral trochlea. Maybe the authors should add there that it does stand for distal and correspond to the trochlea. The same explanation should be add for the other articular surfaces. I will suggest maybe the authors to make a list of abbreviations in the text or as supplementary information.

The abbreviations for all VOIs were already given in the description of the SOM 6.

-p14 line 36 A reference of the peak forces measured in the ferret is needed here.

Added to the RM.

-p17 line 12: The authors talk about the Tb.Th values with the reduction of the forelimb. What about the trabecular spacing and density as these parameters are sometimes correlated? Did the authors find any signals in these last parameters with the forelimb reduction?

Following this suggestion, we have inspected trabecular spacing and density. Tb.Sp and Conn.D follow the same trend as Tb.Th (but of course with an inverse relationship). This is reported in the RM.

Sincerely yours, and on behalf of both co-authors,

Eli Amson

Cited references

Tsegai, Z. J., Skinner, M. M., Pahr, D. H., Hublin, J.-J., & Kivell, T. L. (2018). Systemic patterns of trabecular bone across the human

and chimpanzee skeleton. *Journal of Anatomy*, 232(4), 641–656.